# Selective and stable CO₂ electroreduction at high rates via control of local H₂O/CO₂ ratio

Junmei Chen[1,5], Haoran Qiu[1,2,5], Yilin Zhao [1,5], Haozhou Yang [1], Lei Fan [1], Zhihe Liu[1], ShiBo Xi [3], Guangtai Zheng [1], Jiayi Chen[1], Lei Chen[1], Ya Liu [2], Liejin Guo [2] & Lei Wang [1,4] ✉

Controlling the concentrations of H₂O and CO₂ at the reaction interface is crucial for achieving efficient electrochemical CO₂ reduction. However, precise control of these variables during catalysis remains challenging, and the underlying mechanisms are not fully understood. Herein, guided by a multiphysics model, we demonstrate that tuning the local H₂O/CO₂ concentrations is achievable by thin polymer coatings on the catalyst surface. Beyond the often-explored hydrophobicity, polymer properties of gas permeability and water-uptake ability are even more critical for this purpose. With these insights, we achieve CO₂ reduction on copper with Faradaic efficiency exceeding 87% towards multi-carbon products at a high current density of −2 A cm⁻². Encouraging cathodic energy efficiency (>50%) is also observed at this high current density due to the substantially reduced cathodic potential. Additionally, we demonstrate stable CO₂ reduction for over 150 h at practically relevant current densities owning to the robust reaction interface. Moreover, this strategy has been extended to membrane electrode assemblies and other catalysts for CO₂ reduction. Our findings underscore the significance of fine-tuning the local H₂O/CO₂ balance for future CO₂ reduction applications.

Electrocatalytic CO₂ reduction (CO₂R) coupled with renewable electricity has been identified as an attractive route to produce fuels and chemicals without carbon footprint[1–3]. Previously, numerous catalysts have been developed to direct the selectivity of CO₂R towards valuable fuels/chemicals, including carbon monoxide (CO), formate, and especially multi-carbon products (C₂₊) such as ethylene, ethanol, and propanol[4–7]. However, the major obstacle to the practical implementation of CO₂R remains to achieve sufficient energy efficiency (EE) at high production rates (i.e., >−1 A cm⁻²)[8–10]. In addition to tackling this challenge by searching for new catalysts with improved CO₂R activity and selectivity[11–15], designing and fine tuning the reaction microenvironments are also critical especially for CO₂R under practical conditions[16–18]. This microenvironment refers to the local environment surrounding the catalytic center, including the electrode/electrolyte interfacial structures, local pH, CO₂ and H₂O concentrations, ions/cations if involved, electrical field, etc. All of these factors can greatly influence the kinetics and thermodynamics of the catalytic processes[19,20], thus affecting the conversion efficiency, reaction rate, selectivity, stability and ultimately the energy efficiency of CO₂R[17,21–23].

H-type cells have been extensively employed to evaluate the reaction mechanisms in CO₂R. However, CO₂R current density in H-cells is largely limited (tens of mA per cm²) by the sluggish CO₂ mass transportation due to the low solubility and long diffusion length (~100 μm) of CO₂ in the aqueous electrolyte[24–26]. In contrast, gas-diffusion layer (GDL) based reactors can significantly reduce the thickness of the CO₂ diffusion layer (down to nm scale), boosting the

[1]Department of Chemical and Biomolecular Engineering, National University of Singapore, Engineering Drive 4, Singapore 117585, Singapore. [2]International Research Center for Renewable Energy, State Key Laboratory of Multiphase Flow in Power Engineering, Xi'an Jiaotong University, Xi'an, Shaanxi 710049, China. [3]Institute of Sustainability for Chemicals, Energy & Environment, A*STAR, 1 Pesek Rd, 627833 Singapore, Singapore. [4]Centre for Hydrogen Innovations, National University of Singapore, 1 Engineering Drive 3, 117585 Singapore, Singapore. [5]These authors contributed equally: Junmei Chen, Haoran Qiu, Yilin Zhao. ✉e-mail: wanglei8@nus.edu.sg

maximum $CO_2R$ current density by one to a few orders of magnitude[27,28]. Thus, it is desirable to conduct $CO_2R$ using GDL-based reactors to achieve high energy efficiency at high rates. However, new challenges are associated with GDL-based flow cell. Particularly, the rapid flooding of GDL which can lead to reduced $CO_2R$ selectivity, large concentration overpotentials and system failure, frustrating both lab-scale experiments and scale-up prospects especially at high current densities[29–31].

A handful of strategies have been developed to alleviate the detrimental flooding in $CO_2R$, including pulsed voltammetry[32], electrolyte refreshing[33], functionalizing the GDL with a fluorocarbon silane/Polytetrafluoroethylene (PTFE)[4,34], replacing the carbon-based substrate with PTFE film[35], and increasing the hydrophobicity of the catalyst layer[36–38], etc. In general, these strategies aim to physically remove the salt precipitations and/or enhance the hydrophobicity of the GDL to prevent flooding. Among these methods, incorporating hydrophobic polymers into the catalyst layer has been proven to be effective in increasing resistance to flooding. It is believed that the hydrophobic backbones or pores within these additives or polymers trap more $CO_2$ near the catalyst surface and repeal water, thereby regulating local $CO_2$ and $H_2O$ concentrations[37,38]. This regulation subsequently results in increased current density and selectivity towards $CO_2R$ products[34,38,39]. However, it is challenging to attribute this enhancement solely to hydrophobicity, considering other factors such as polymer porosity or catalyst morphology, and valence state changes induced by the introduced polymer. Additionally, the specific mechanism by which hydrophobic polymers affect product selectivity and activity, especially for multi-hydrocarbon products, remains a subject of controversy. Hence, a thorough understanding and investigation of hydrophobicity's influence in the $CO_2R$ process is crucial. Moreover, although hydrophobicity modification of the gas diffusion electrode (GDE) has promoted the current of $CO_2R$, achieving an industrially significant current density beyond $-1\,A\,cm^{-2}$ with high $CO_2R$ selectivity remains a challenge.

In this study, we aim to uncover the origin of the hydrophobicity effect on the formation of multi-hydrocarbon products ($C_{2+}$) regarding selectivity, activity and stability, and employ the new insights gained to achieve selective and energy efficient $CO_2R$ at industrial relevant current density. Often, the hydrophobicity of the catalyst layer was regulated by introducing hydrophobic polymers, such as PTFE powders[36]. While improved Faradaic efficiency (FE) of $C_{2+}$ products were obtained, abruptly decrease in FE of $C_{2+}$ products still occurred when the current density exceeded $-0.5\,A\,cm^{-2}$ (Supplementary Fig. 1) due to severe GDL flooding. We hypothesize that the origin of the instability of this design is the short-ranged effect of PTFE on

modulating the local $CO_2$ and $H_2O$ concentrations, since the large PTFE powders were simply mixed with the catalyst particles physically (Supplementary Fig. 2), as illustrated in Model I in Fig. 1. In contrast, we propose that a thin and intact polymer layer on the catalyst surface (Model II in Fig. 1) would be more effective in preventing the GDL flooding while preserving the catalytic performance, provided that this thin layer can effectively manage the transportation of $CO_2$, $H_2O$ and $CO_2R$ products. Specifically, the polymer layer thickness, water uptake ability, $CO_2$ diffusivity, porosity and other related chemical/physical properties will provide us opportunities in optimizing the micro-environment (i.e., $H_2O/CO_2$ balance) at the triple-phase interface, enabling selective and energy efficient $CO_2R$ at high rates.

Building on the above considerations, we identified four polymers (PCR: Poly[3,3,4,4-tetrafluoro-2-methyl-2-(1,1,2,2,3-pentafluoro-3-(trifluoromethoxy)butyl)−5,5-bis(perfluoroethyl)tetrahydrofuran]; PT95: Poly[2-(1,1-difluoroethyl)−2-ethyl-4,4,5,5,6-pentafluoro-6-(trifluoromethyl)−1,3-dioxane]; PVDF: Polyvinylidene fluoride, and Nafion 117, Fig. 2) with varying degrees of hydrophobicity for preparing Cu GDEs according to mode II. When assessing these Cu GDEs under identical conditions, we found that their $CO_2R$ performance did not scale with the hydrophobicity of the corresponding polymer (Fig. 2). This indicates that hydrophobicity is not a direct descriptor for the $CO_2R$ activity and selectivity towards multi-carbon products. To pinpoint the more relevant factors, we conducted Multiphysics modeling based on Mode II-type catalyst layer, in which we introduced a thin polymer layer onto the catalyst surface. We discovered that the $CO_2$ to $H_2O$ ratio at the catalytic interface, which predominately determined by both the hydrophobicity and the porosity of the polymer, plays a critical role in the controlling the $CO_2R$ activity and selectivity. Furthermore, if the thin polymer layer is chemically stable and can remain intact during $CO_2R$, we can anticipate a GDE with exceptional resistance to flooding. Accordingly, we identified a fully perfluorinated polymer (PT: Poly[4,5-difluoro-2,2-bis(trifluoromethyl)−1,3-dioxole-co-1,3-Dioxane,2-(difluoromethylene)−4,4,5,5,6-pentafluoro-6-(trifluoromethyl)]) and successfully prepared a Model II-type GDE, thanks to its desirable properties such as great chemical stability, decent solubility in common solvents, high $CO_2$ diffusivity, high hydrophobicity, suitable water uptake ability and porosity, etc, it enabled an optimized and robust local $H_2O/CO_2$ ratio during $CO_2R$. As a result, we obtained a high FE of over 87% for $C_{2+}$ products at an exceptionally high current density of $-2\,A\,cm^{-2}$ using PT/Cu-based GDL. This represents a two-to-three-fold improvement compared to the maximum $C_{2+}$ partial current densities obtained from other four polymer/Cu electrodes. Notably, an over 150 h and 10 h stable $CO_2R$ electrolysis were achieved using PT/Cu at $-0.2\,A\,cm^{-2}$ and $-1.0\,A\,cm^{-2}$, respectively, with negligible losses in both activity and selectivity. In contrast, the $CO_2R$ on Nafion/Cu failed rapidly, especially under high current densities. Furthermore, the optimized microenvironment led to significantly reduced overpotential for $CO_2R$ on PT/Cu compared to those on other four polymer/Cu electrodes (i.e., by $>100\,mV$ at $-1.5\,A\,cm^{-2}$ compared to PCR/Cu). As a result, high cathodic energy efficiency (EE) exceeding 50% towards $C_{2+}$ products were achieved on PT/Cu at high current density of $-1.5\,A\,cm^{-2}$. Furthermore, we evaluated this strategy for $CO_2R$ in acidic electrolyte and in membrane electrode assemblies (MEA) based reactors to minimize the carbonate formation. Encouragingly, compared to Nafion/Cu, the overpotentials (in acidic electrolyte) and cell voltage (in MEA-reactor) of PT/Cu-based $CO_2R$ at $-1\,A\,cm^{-2}$ was reduced substantially by 0.4 V and 0.3 V, respectively. Lastly, the same concept has been successfully extended to Sn-based and Ag-based $CO_2R$ systems to produce formate and CO, respectively. Overall, this work established a facile and effective strategy for constructing catalyst layers that markedly enhance the stability and energy efficiency of practical relevant $CO_2R$. More importantly, the insights gained regarding the fine-tuning of the

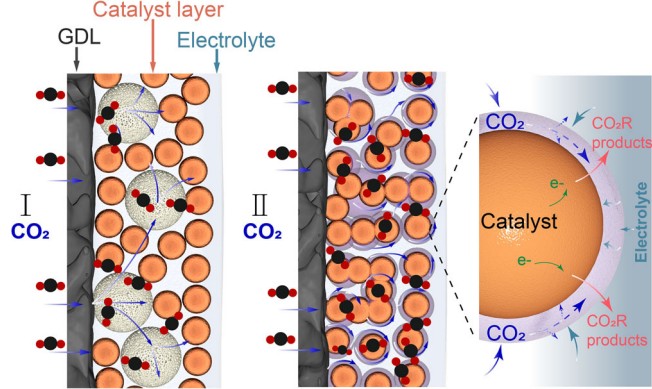

**Fig. 1 | Schematic of the two modes in managing the $CO_2/H_2O$ balance at the $CO_2R$ reaction interface.** Orange sphere: catalyst particle; white sphere in mode I: PTFE powder; purple layer in mode II: thin and intact polymer coatings with optimized $CO_2$ and $H_2O$ management properties.

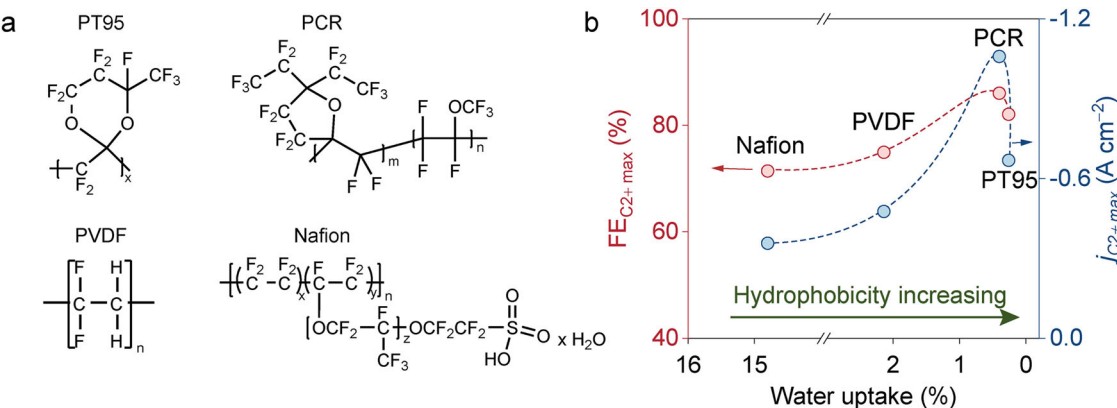

**Fig. 2 | Chemical structures of the polymers and their CO₂R performances.**
**a** Chemical structures of the four hydrophobic polymers. **b** correlation between polymer hydrophobicity and the performance of C₂₊ products from CO₂R.

Commercial Cu powder was used as the model catalyst. Relevant source data are provided as a Source Data file.

water/gas balance at the reaction interface hold significance for the future development of CO₂R electrolyzers.

## Results

### Investigating the impact of hydrophobicity on CO₂R performance

To investigate the impact of hydrophobicity on CO₂R performances, we selected four hydrophobic polymers to construct the Model II-type catalyst layer. The chemical structures of these four polymers are illustrated in Fig. 2a. PT95 and PCR are both fully fluorinated polymers with high hydrophobicity. Polyvinylidene fluoride (PVDF) was also selected due to its similar structure with PTFE, however, it possesses the advantage of being soluble in organic solvents such as dimethylformamide and N-methyl-2-pyrrolidone[40], which is critical for preparing thin film coatings on Cu particles. Additionally, we employed Nafion as the reference sample since it is widely used for preparing the catalyst layer for CO₂R[41,42]. We employed commercial Cu nanoparticles (Cu, ~25 nm) as the model catalyst. Although this Cu is less active/selective than specifically designed Cu catalysts[28,43] it has been extensively used as a benchmark catalyst for CO₂R, making our conclusions more instructive. Conventional GDL preparation procedures were employed to prepare the polymer/Cu catalyst. Specifically, PCR, PT95, PVDF and Nafion were mixed with Cu in solvents capable of dissolving the polymers, followed by thorough sonication to afford homogenous catalyst ink solutions. This solution was then spray-coated to form the Cu catalyst layer on GDL. While these polymers were used as catalyst binders, our design objective also aimed for thin and uniform coatings on the catalyst surface.

First, we assessed the relative hydrophobicity of the four polymer/Cu GDEs using contact angle measurements. As depicted in Supplementary Fig. 3, all four GDEs displayed excellent hydrophobicity, with contact angles greater than 140°. Note that we were not able to capture the conventional contact-angle images for the as-prepared PT95/Cu and PCR/Cu GDEs owing to their super-hydrophobicity (Supplementary Movies 1 and 2). We also examined the water uptake properties of the four polymers to further differentiate their hydrophobic levels. The results indicated that PT95 is likely more hydrophobic than PCR (Supplementary Fig. 4). Nevertheless, we believe the relative initial hydrophobicity of the four polymers is in the sequence of PT95 > PCR > PVDF>Nafion (Fig. 2b).

Then, these polymer/Cu GDEs were evaluated for CO₂R using a flow-cell reactor (Supplementary Fig. 5). Using the obtained data, we plotted the highest C₂₊ products selectivity and partial current density achieved by the four polymer/Cu GDEs against their hydrophobicity. As shown in Fig. 2b, we found that both the C₂₊ selectivity and current density initially scale with the increased hydrophobicity, however, they decline after reaching a certain region. This trend suggests that, while the hydrophobicity of the polymer coating plays a role in enhancing CO₂R as suggested elsewhere[37,38,44], it is not the sole contributor. Consequently, other variables such as polymer porosity and layer thickness, water uptake ability, CO₂ diffusivity, and other related chemical/physical properties must be considered. Through studying and adjusting these parameters, we believe the microenvironment can be further optimized towards efficient CO₂ to C₂₊ conversion.

### Modeling the mass balance in Model II

We developed a direct pore-level multi-physics model (Fig. 3a, Supplementary Figs. 6 and 7, details provided in Supplementary Information) to simulate the distribution of species on the catalyst surface during CO₂R, and to evaluate the CO₂R performance while changing the relevant properties of the polymer coating, i.e., thickness, porosity, and ability in managing the water/gas balance at the catalyst surface. In this model, the simulation region was limited to a single pore within the catalyst layer, and a thin/porous polymer layer was employed between the catalyst surface and the electrolyte domain. The thickness of this polymer layer is defined as $Thk_{PL}$. The pore width of the catalyst layer is assumed as 200 nm[35]. The span of the simulation space equates to the thickness of the catalyst layer (~30 μm). To conserve computational resources, the liquid electrolyte domain is conceived to be half-sized at 100 nm, complemented with symmetrical boundary conditions. Besides, both the interface mass transfer of CO₂ and the mass transfer of the bulk electrolyte are both characterized by mass flux boundary conditions, with a constant rate for the interface mass transfer of CO₂ and a consistent composition of the bulk electrolyte throughout the simulation, as illustrated in Fig. 3a[45,46]. Transport and electrochemical kinetics are resolved in this model to investigate how the CO₂R is affected by different properties of the polymer coating. As illustrated in Fig. 1, we hypothesize that the transition from Model I to II would immediately alter the balance of H₂O and CO₂ at the reaction layer. Thus, we chose the volume ratio of H₂O to CO₂ (H₂O/CO₂) near the catalyst surface as one key variable of our modeling. Additionally, both porosity and thickness of the polymer layer were taken into consideration in our model to assess how they affect the local microenvironment (local species concentration) and further the CO₂R performance[47,48]. We initially assessed the impact of polymer thickness and found that it had relatively small effects on the local microenvironments (i.e., CO₂ concentration, pH) and the CO₂R performance, as shown in Supplementary Fig. 8. This limited effect observed in our simulations, where the polymer thickness ranges from 0 to 30 nm, can be attributed to the fact that this thickness range does not

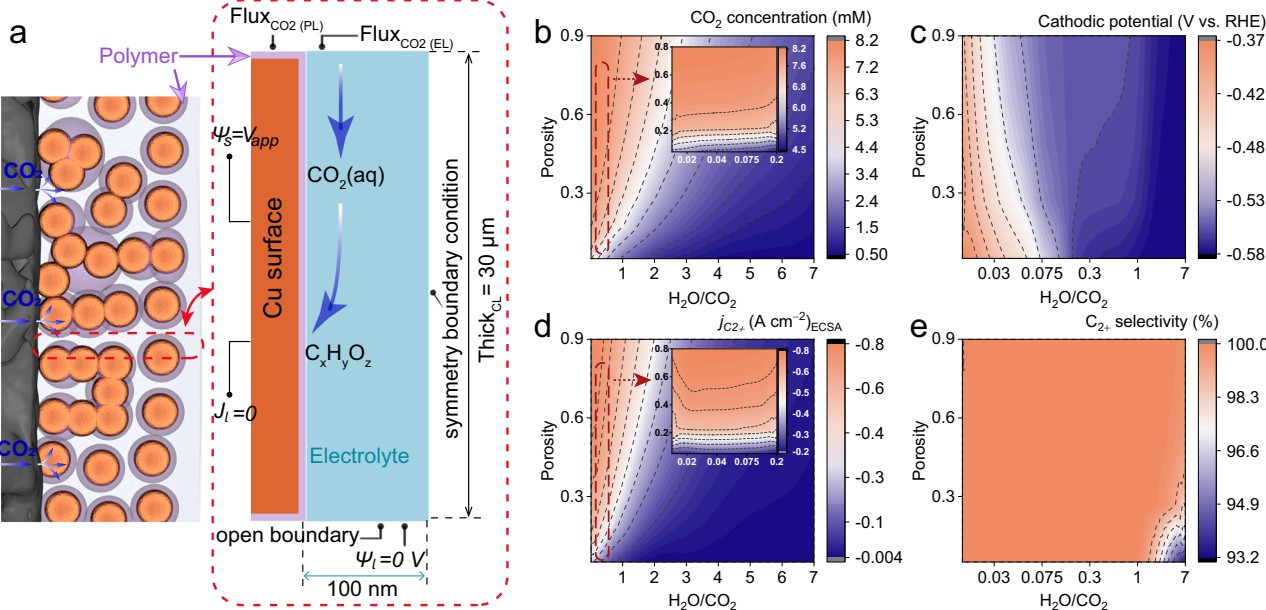

**Fig. 3 | Multi-physics Modeling of $CO_2$R in Model II. a** Scheme of $CO_2$ and $H_2O$ transport in the polymer modified catalyst (left) and the graphical illustration of the modeling domains (right). The gas and liquid transport are decoupled, leading to a three-phase unsaturated layer for enhanced electron participation in desired electrochemical reaction and reduced the $CO_2$ transport length to the catalyst surface. The source of protons for $CO_2$R is the water taken up by the polymer coating. **b** Modeled $CO_2$ availability for the desired $CO_2$R. **c** Cathodic potential. **d** $C_{2+}$ products current density and **e** $C_{2+}$ selectivity with the variation of the polymer porosity and local $H_2O/CO_2$ ratio at the same applied cathodic potential (−1.426 V vs. SHE). The inserts in panels **b** and **d** are the enlarged areas indicated by the red dashed line. $CO_2$ flux boundary conditions are set at the upper boundaries

of both the liquid electrolyte domain ($Flux_{CO_2(EL)}$) and the polymer domain ($Flux_{CO_2(PL)}$). A symmetrical condition is imposed at the right boundary to model a confined pore geometry in the catalyst layer (CL). The electrolyte potential $\psi_l$ at this boundary is set to 0 V to provide a reference for solving the electric field. Due to the continuous flow of fresh 2 M KOH electrolyte into the cathode chamber, an open boundary condition is imposed at the bottom boundary, and the equilibrium concentration values of $K^+$ ($c_{K^+}^{eq}$) and $OH^-$ ($c_{OH^-}^{eq}$) are set at this boundary (Supplementary Table 1-3). A fixed cathodic potential $\psi_s = V_{app}$ and electrolyte current density $J_l = 0$ are imposed at the left boundary. Relevant source data are provided as a Source Data file.

significantly influence the effective diffusion coefficients of species within the domain. Consequently, $CO_2$ reactants and other species within the electrolyte diffuse quickly through the polymer layer, at least at rates faster than the reaction kinetics. Besides, given the significantly higher concentration of gaseous $CO_2$ compared to dissolved $CO_2$ in the electrolyte, and its diffusion coefficient being approximately 10,000 times greater[49], the impact of the polymer layer thickness on $CO_2$ mass transfer becomes relatively small. Separately, we did not consider a much thicker polymer layer in our model because increasing the porous layer thickness would compromise electrode conductivity. Notably, we did not consider the impact of conductivity changes (which are difficult to reflect in the simulation) when considering variations in thickness in the simulation. By contrast, the thin layer porosity influences the microenvironments and $CO_2$R performance obviously, as it directly affects the effective diffusion coefficient of all species within the layer. Based on our results, we suggest that the desired porosity range for polymer materials should be ranging from 0.2 to 0.9 (Fig. 3b-e)[50,51]. If the polymer porosity is too low, the diffusion of $CO_2$ and other species will be significantly limited, thereby impeding the $CO_2$R.

As shown in Fig. 3b, it is apparent that small $H_2O/CO_2$ ratio and high polymer porosity enable high local $CO_2$ mass transfer efficiency at the catalyst surface (Supplementary Fig. 9), resulting in enhanced $CO_2$R partial current density at consistent applied cathodic potentials (vs. SHE, Supplementary Fig. 10a). Besides, the local pH increases monotonically with the decrease of the $H_2O/CO_2$ ratio (Supplementary Fig. 10b). This is expected because the total amount of $OH^-$ generated from $CO_2$R increases with higher current densities at given cathodic potentials. Notably, the $OH^-$ concentration change is more pronounced in regions with low $H_2O/CO_2$ ratio (<0.3). Previous work has

suggested that high $OH^-$ concentration could lower the activation barriers of CO-CO coupling[28]. Thus, it may also contribute to the observed reduced cathodic potential in the low $H_2O/CO_2$ ratio region (Fig. 3c). The overall high local $CO_2$ concentrations also favor the production of $C_{2+}$ products (Fig. 3d). However, the $C_{2+}$ selectivity is less sensitive to these two variables (Fig. 3e). Indeed, on one hand, the decreased cathodic potentials at a low $H_2O/CO_2$ ratio corresponding to a decrease in $C_{2+}$ products selectivity, similar potential dependent trend in $CO_2$R selectivity observed previously[52,53]. On the other hand, the $C_{2+}$ selectivity might also be influenced by the local $CO_2$ concentration, as higher population of unreacted $^*CO_2$ may compete with CO dimerization for the Cu-sites. The combined effects of lower cathodic potential and higher local $CO_2$ concentration retain the $C_{2+}$ selectivity despite various in the $H_2O/CO_2$ ratio and porosity. As the pore size of the catalyst layer can affect the effective diffusion of $CO_2$ gas[47], we explored the changes in the $CO_2$R performance when the catalyst layer pore size ranged from 10 nm to 100 nm[35] (Supplementary Figs. 11 and 12). First, we simulated the local $CO_2$ concentration as the pore size increased, however, with a relatively modest increase of approximately 1.5 mM when the pore size increased tenfold, from 10 nm to 100 nm. Consequently, the cathodic current density shows a minor decrease as the pore size decreases from 100 nm to 40 nm. Notably, we observed a sharp decline in the current density when we further reduced the pore size to 10 nm, indicating that significant $CO_2$ mass transportation limitation occurred at this scale. Furthermore, the changes in both the local pH (Supplementary Fig. 11c) and cathodic potential (Supplementary Fig. 11d) as a function of pore size show a similar trend to what was observed for the cathodic current density, indicating that the effect of the pore sizes on the $CO_2$R is minimal unless a small pore size of <10 nm predominates in the catalyst.

Overall, these simulations suggest that regulating the polymer porosity and the local $H_2O/CO_2$ ratio are effective approaches for optimizing the microenvironment and enhancing the $CO_2R$ performance.

## Tuning the $H_2O/CO_2$ ratio at the surface of Cu catalyst

Motivated by the above simulation results, we sought to find a suitable polymer that could enable us to construct the Model II-type catalyst layer and afford an optimized microenvironment for efficient $CO_2R$. The ideal polymer candidate should be hydrophobic and possess suitable porosity to manage an optimized local $H_2O/CO_2$ ratio (determined by both the water uptake ability and $CO_2$ adsorption capacity). Additionally, good chemical stability, especially under practical $CO_2R$ conditions, is also critical.

As shown in Fig. 4a, a fully perfluorinated polymer resin (PT) was selected due to its high hydrophobicity, $CO_2$ adsorption ability, high porosity (Supplementary Table 3), and excellent chemical stability. First, we measured the $CO_2$ adsorption capacities of PT and the above-mentioned polymers (PT95, PCR, PVDF, Nafion) using $CO_2$ adsorption isotherms. As shown in Fig. 4b and Supplementary Fig. 13, PT exhibits significantly higher $CO_2$ adsorption compared to the other polymers. Note that the $CO_2$ adsorption measurements were conducted under dry conditions, thus, it is expected that the $CO_2$ adsorption will be reduced if severe water uptake occurs, i.e., during flooding. Then, we measured the water uptake ability of these polymers (Methods and Supplementary Fig. 14). As depicted in Fig. 4b, under identical conditions, PT absorbed only 0.58 wt% of water, which is approximately four and twenty-four times lesser than that of the PVDF and Nafion, respectively. This low water uptake ability of PT is comparable to those of PCR and PT95 (Supplementary Tables 4 and 5). Taken together, we tentatively estimated the $H_2O/CO_2$ ratio based on the polymer loadings (Supplementary Table 6) and their corresponding water and $CO_2$ uptakes. It was observed that the PT polymer led to the lowest $H_2O/CO_2$ ratio at the catalyst surface, which is 10 and 658 times lower than that of PVDF and Nafion, respectively (Fig. 4c, Supplementary Table 3). Notably, PT also exhibits a lower local $H_2O/CO_2$ ratio compared to both PCR and PT95, even though they are more hydrophobic than PT. We believe that the low $H_2O/CO_2$ ratio of PT will lead to further improved $CO_2R$ performance. To demonstrate this assumption, we prepared the Model II-type PT/Cu GDE following the above procedure and conducted comprehensive $CO_2R$ measurements.

## Characterizations of the Model II-type Cu catalyst layer

Prior to the GDL preparation, X-ray photoelectron spectroscopy (XPS) was performed to study the composition of the polymer modified Cu. As shown in Fig. 4d, all polymer-modified Cu possess an F $1s$ peak, confirming the surface coating of polymers. The chemical state of Cu was then investigated using X-ray absorption spectroscopy (XAS). The acquired Cu K-edge XAS spectra for the five samples were identical (Supplementary Fig. 15), indicating that the chemical state of the Cu catalyst was not altered by the polymer coating. The Cu LMM Auger spectra (Supplementary Fig. 16) further confirm this. Additionally, the X-ray diffraction (XRD) patterns confirm that the crystalline structure of the Cu NPs was retained during the polymer coating (Supplementary Fig. 17). Note that a small fraction of diffraction peaks for $Cu_2O$ were observed in the XRD patterns, which was probably caused by air exposure. The presence of $Cu_2O$ is further supported by the XAS results, which indicate mixed valence states in the precursor Cu NP (Supplementary Fig. 15)[36].

The morphology of the polymer-coated Cu was examined using Transmission Electron Microscopy (TEM). As shown in Fig. 4e–i and Supplementary Fig. 18, comparable morphologies were found for all five coated Cu NPs, which are consistent with the Scanning Electron Microscopy (SEM) images (Supplementary Fig. 19). Additionally, the uniform thin layers (marked by red arrows) on the Cu surfaces indicate that the polymer coatings are evenly distributed across the catalyst surface. The thicknesses of these polymer layers were determined by the high revolution TEM (HRTEM) (Fig. 4f–j, Supplementary 17b–f and Supplementary Fig. 20). With the typical polymer loadings, average coating thickness of 3.65 nm, 4.19 nm, 4.37 nm, 4.75 nm and 4.19 nm were determined for PT, PVDF, Nafion, PCR and PT95, respectively. The catalyst-loading in each electrode was approximately 1 mg cm$^{-2}$, and the polymer loading could then be determined (Supplementary Table 6).

The cross-sections of the polymer/Cu GDEs were characterized by SEM. As shown in Fig. 4k–p and Supplementary Fig. 18g–l, no significant differences were observed in the catalyst-layer thickness and the structures of the microporous layer across the three samples, indicating the similar electrode geometry for the polymer-modified GDEs. However, a slightly denser catalyst-layer was observed for PT/Cu, PCR/Cu and PT95/Cu (i.e., in absence of obvious cracking), as illustrated in Fig. 4p compared to Supplementary Fig. 18h–l. This difference could be attributed to a more uniform dispersion of the PT/Cu, PCR/Cu and PT95/Cu catalyst-inks, resulting in fewer agglomerates during spray coating. As a result, one could expect that the reactants (i.e., $CO_2$) have less restricted access to the catalyst surface[54]. Overall, we have successfully prepared GDEs with Mode II-type catalyst-layers. While the chemical/physical properties of the Cu catalysts are nearly identical among these GDEs, we anticipate that the different polymer-coatings will lead to distinct $CO_2R$ performances due to the different local environment created, particularly the different local $H_2O/CO_2$ ratios.

## Electrochemical $CO_2R$ on polymer/Cu GDEs in flow cell

To assess the validity of our model and design, $CO_2R$ catalyzed by the above GDEs was evaluated using a flow-cell reactor under identical conditions (Supplementary Fig. 5). Hereinafter, all potentials are referenced to the reversible hydrogen electrode (RHE), unless otherwise noted.

As shown in the polarization curves (Fig. 5a, Supplementary Fig. 21a), PT/Cu exhibited higher apparent activity than other polymer/Cu GDEs under the same conditions, in line with simulated results (Supplementary Fig. 21b). Supplementary Fig. 22 depicts the FEs for all detectable products from the $CO_2R$ with different GDEs. In general, all five GDEs exhibit similar products distributions, suggesting similar reaction pathways involved (Supplementary Tables 7–11). Encouragingly, the PT/Cu GDEs clearly outperforms other GDEs, as evidenced by its significantly reduced HER (< 8% from −0.1 to −2 A cm$^{-2}$) and improved $C_{2+}$ selectivity at high current densities (Fig. 5b, Supplementary Fig. 23). Besides, its $FE_{C2+}$ increased monotonically with the current density and reached 87.4% at −2 A cm$^{-2}$, exceeding those of the state-of-the-art systems under similar conditions[4,35,55,56]. In contrast, other GDEs reached their peak $C_{2+}$ selectivity at relatively lower current densities. Specifically, PCR exhibited an $FE_{C2+}$ of 86% at −1 A cm$^{-2}$, while PT95/Cu, PVDF/Cu and Nafion/Cu achieved peak $FE_{C2+}$ at −0.5 A cm$^{-2}$, followed by rapid deactivation in $CO_2R$ (Fig. 5b). Meanwhile, the corresponding HER activity for these polymer/Cu GEDs increased sharply after reaching their peak $FE_{2+}$. This increase can be attributed, on one hand, to the rapidly increasing of local $H_2O/CO_2$ ratio as the consumption of $CO_2$ and accumulation of $H_2O$ during $CO_2R$ process, on the other hand, to the occurrence of severe flooding, especially in the case of PVDF/Cu and Nafion/Cu. The easy flooding phenomenon of PVDF/Cu and Nafion/Cu will be thoroughly discussed later.

Furthermore, PT/Cu required significantly reduced overpotentials compared to the other four GDEs, particularly at high current densities (Fig. 5c). Consequently, PT/Cu afforded significantly higher partial current densities for $C_{2+}$ production at the same cathodic potential (Fig. 5d). As a result, high cathodic energy efficiency (EE) of over 50% was obtained for PT/Cu catalyzed $C_{2+}$ production at high current density of −2 A cm$^{-2}$ (Supplementary Fig. 24), demonstrating the

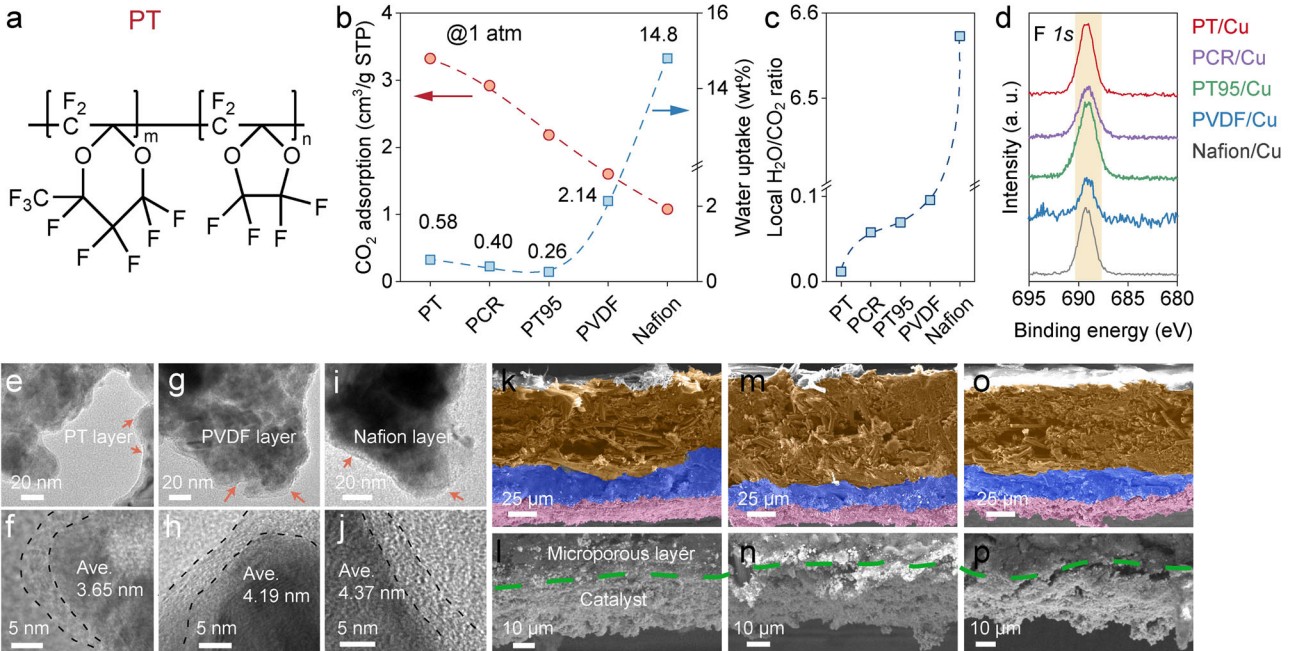

**Fig. 4 | Characterizations of Cu modified by different polymers. a** Chemical structure of PT polymer resin. **b** $CO_2$ adsorption isotherms and water uptake of PT, PT95, PCR, PVDF and Nafion polymer membrane. **c** calculated local $H_2O/CO_2$ ratio based on the polymer loadings and their water uptake and $CO_2$ adsorption ability. **d** F $1s$ XPS spectra of PT, PCR, PT95, PVDF and Nafion. **e–j**, FETEM images of (**e, f**) PT/Cu, (**g, h**) PVDF/Cu and (**i, j**) Nafion/Cu. **k–p** cross-sections of (**k, l**) PT/Cu, (**m, n**) PVDF/Cu and (**o, p**) Nafion/Cu based GDLs. In panels **k–o**, false colors applied to the images for clarity, pink: catalyst layer, blue: microporous layer, orange: gas diffusion layer. The white color on the top of the GDE is the spray coated PTFE. The average polymer thicknesses have been measured at least ten times at different samples/locations, more FETEM images can be found in Supplementary Information. Relevant source data are provided as a Source Data file.

promise of practical implemetations[4,28]. Contrastingly, other polymer/Cu GDEs, namely PCR/Cu, PT95/Cu, PVDF/Cu and Nafion/Cu achieved their highest $EE_{C2+}$ at much lower current densities: $-1\,A\,cm^{-2}$ for PCR/Cu and $-0.5\,A\,cm^{-2}$ for the other three polymer/Cu GEDs. We correlated the highest $FE_{C2+}$ and $j_{C2+}$ achieved by these five polymer/Cu GDEs with the corresponding estimated $H_2O/CO_2$ ratio (Supplementary Fig. 25). A clear scaling trend was observed, highlighting the significance of attaining and sustaining a low local $H_2O/CO_2$ for achieving selective and stable $CO_2R$ at high rate with enhanced energy efficiency. Also, the observed reduction in cathodic potentials aligns with the simulation results (Fig. 3c, d). To exclude the possibility of over-compensation of $iR$, particularly at high current densities, non-compensate overpotentials were compared among these samples (Supplementary Fig. 26), and the same trends were observed. Worth noting that, although PT95 and PCR exhibit similar local $H_2O/CO_2$ ratios (PCR: 0.06, PT95: 0.07, Supplementary Table 3), their $CO_2R$ performances differ significantly. This discrepancy may be attributed to the low porosity of both polymers (<0.2 with PT95 at 0.07, PCR at 0.14). Such low porosity hinders the diffusion of local species (e.g. $CO_2$, $H_2O$, $OH^-$), and substantially reducing local $CO_2$ concentration, and negatively impacting the $CO_2R$ performances (Fig. 2b–d). Overall, polymers capable of regulating lower local $H_2O/CO_2$ ratios can facilitate efficient $CO_2R$ at higher current densities and increase energy efficiency.

On the other hand, PT/Cu exhibits relatively low $C_{2+}$ selectivity but higher CO selectivity compared to other polymer/Cu GDEs at low current densities, i.e., $\leq -0.5\,A\,cm^{-2}$ (Fig. 5b, Supplementary Fig. 27a, Supplementary Table 12). This can be attributed to the lower cathodic potentials applied (Fig. 5d, Supplementary Fig. 27b), in agreement with our modeling results and the previously reported potential-dependent selectivity trends for $CO_2R$[52,53]. To further analyze the kinetics, we examined the partial current densities of $C_{2+}$, CO and $H_2$ for these GDEs (Fig. 5d, e and Supplementary Fig. 28). As anticipated, the enhanced $C_{2+}$ and CO partial currents were associated with the decrease in local

$H_2O/CO_2$ ratio regulated by the polymers. However, the $H_2$ partial current densities and their trends were close for all five GDEs, as shown in Fig. 5e, suggesting that the HER activity is not directly associated with the local $H_2O/CO_2$ ratio. We then assessed the electrochemical active surface area (ECSA) of the five GDEs. While we acknowledge that the simplified ECSA measurements may not fully reveal the exact picture of the reaction interface during $CO_2R$, we believe that the minor differences in their ECSA are likely not the primary factor causing the difference in their $CO_2R$ performance (Supplementary Fig. 29). We further plotted the selectivity of $C_{2+}$ products and CO for each polymer/Cu electrode against their potentials, as illustrated in Fig. 5f and Supplementary Fig. 27b. Interestingly, in the non/minor-flooded region (<$-0.65\,V$), the selectivity towards $C_{2+}$ and CO are similar across all these polymer/Cu electrodes. However, further increase the cathodic potential tends to induce flooding in all polymer/Cu GDEs[29]. PT demonstrated the best flooding resistance, as evidenced by the continued increase in $FE_{C2+}$. Altogether, we believe that a polymer coating capable of regulating a low local $H_2O/CO_2$ ratio will lead to an optimized $CO_2R$ microenvironment, minimizing concentration overpotential and allowing for improved $CO_2R$ current density. Also, the polymer coating will not significantly affect product selectivity at the same applied cathodic potential within the non/minor-flooded region.

Next, we investigated the relative stability of these polymers by measuring the water and gas contact angles of the post-electrolysis electrodes. As shown in Fig. 6a, all five GDEs displayed excellent hydrophobicity prior to $CO_2R$, with water contact angles great than 140°. Note that we were unable to capture the conventional contact-angle images for the as-prepared PT/Cu GDE owing to its super-hydrophobicity (Supplementary Movie 3). Nevertheless, PT/Cu maintained consistently high-water contact angles of over 150° throughout the one hour $CO_2R$ at $-0.5\,A\,cm^{-2}$. PCR/Cu and PT95 also exhibited commendable hydrophobic stability during the test, though they were slightly less stable than PT/Cu. In contrast, both PVDF/Cu and Nafion/

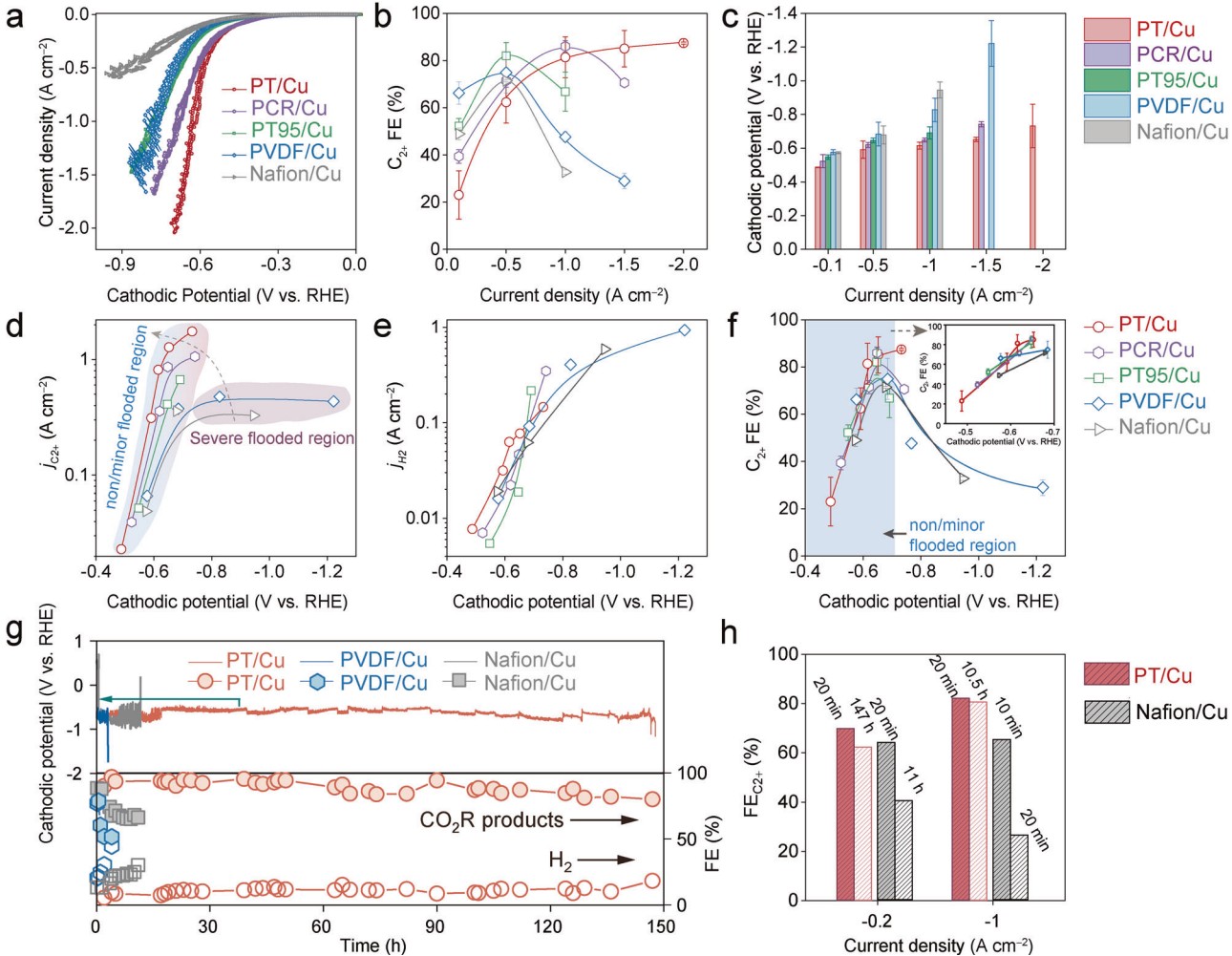

**Fig. 5 | Electrochemical CO₂R on different polymer/Cu GDEs. a** Polarization curves of the different polymer/Cu GDEs (with 85% *iR* compensation. The solution resistance for PT/Cu, PCR/Cu, PT95/Cu, PVDF/Cu and Nafion/Cu are 0.31 Ω, 0.35 Ω, 0.33 Ω, 0.31 Ω and 0.38 Ω, respectively). **b** C₂₊ FE and **c** overpotential on these polymer/Cu GDEs under the same current density. **d** C₂₊ partial current density. **e** H₂ partial current density and **f** C₂₊ FE on these polymer/Cu GDEs under identical cathodic potentials. **g** stability of PT/Cu, Nafion/Cu and PVDF/Cu at −0.2 A cm⁻² in 1 M KOH. **h** C₂₊ selectivity of PT/Cu and Nafion/Cu before and after stability test at −0.2 A cm⁻² and −1 A cm⁻² with 1 M KOH as the electrolyte. The stability tests were

stopped when H₂ FE higher than 15%. For panels **a**–**f**, the test conditions include 2 M KOH as the electrolyte, a CO₂ flow rate of 24 sccm, and a catholyte flow rate of 2 mL/min. For all the tests in a flow cell, the anolyte flowrate is 5 mL/min. Regarding the cathodic potential in panel **a**, 85% *iR* correction was applied. For panels **c**–**g**, 100% *iR* correction was applied. Details regarding the *iR* correction can be found in the Method section. The solution resistances for all measurements in panels **b**–**f** were recorded and plotted in Supplementary Fig. 26a. The error bars in panel **b**, **c** and **f** represent standard deviations from at least three independent measurements. Relevant source data are provided as a Source Data file.

Cu showed a rapid decrease in water contact angles under the same conditions (Supplementary Fig. 30). Particularly, the hydrophobicity of PVDF was significantly compromised within the first 10 minutes of electrolysis (Supplementary Movie 4). This decrease in hydrophobicity is closely associated with the electrode-flooding during CO₂R. When flooding occurs, the GDL draws electrolyte from the reaction interface, resulting in continuous salt-precipitation. This lead to the blockage of CO₂ transport-channels[57] and consequently substantially increased concentration overpotential for CO₂R. The change in hydrophobicity of these electrodes can be explained by electrowetting. To mimic the electrowetting effect under in-situ test conditions, we evaluated the variation in contact angle for PT/Cu and Nafion/Cu by employing a custom-designed in situ water contact angle assessing platform. As shown in Supplementary Fig. 31, the droplet contact angle on PT/Cu exhibited a smaller change compared to that on the Nafion/Cu electrode. This observation further confirms the enhanced hydrophobic stability provided by PT modification.

We also designed and conducted gas contact angles measurements for these GDEs to evaluate their catalyst layer gas affinity, mimicking the

reaction conditions, i.e., in presence of aqueous electrolyte. After CO₂R electrolysis at −0.5 A cm⁻² for 30 min, both Nafion and PVDF displayed significantly large gas contact angles (Supplementary Fig. 32), signifying strong gas repellence and their diminished hydrophobicity (Fig. 6a). PCR, however, show a contact angle of just 23° after CO₂R, suggesting a much better gas affinity. Both PT and PT95 remain good gas affinity, as the gas bubbles were quickly absorbed by the surface (Supplementary Movies 5 and 6). Furthermore, even when the testing durations and currents were extended for PT/Cu, it still maintained superior gas affinity (Supplementary Fig. 33). Comparing the stability of water hydrophobicity and gas affinity among these GDEs, PT clearly outperformed the other polymers in terms of stability across both time and current densities scales. This characteristic ensured its robustness towards flooding and enhanced CO₂R reaction rate. While for PVDF, although it initially showed a low H₂O/CO₂ ratio, its rapidly decreased water hydrophobicity and gas affinity likely led to substantially increase in local H₂O/CO₂ ratio, negatively impacting CO₂R performance.

PT's stable hydrophobicity was further substantiated by the elemental mapping on the cross-section of the GDEs post CO₂R. We

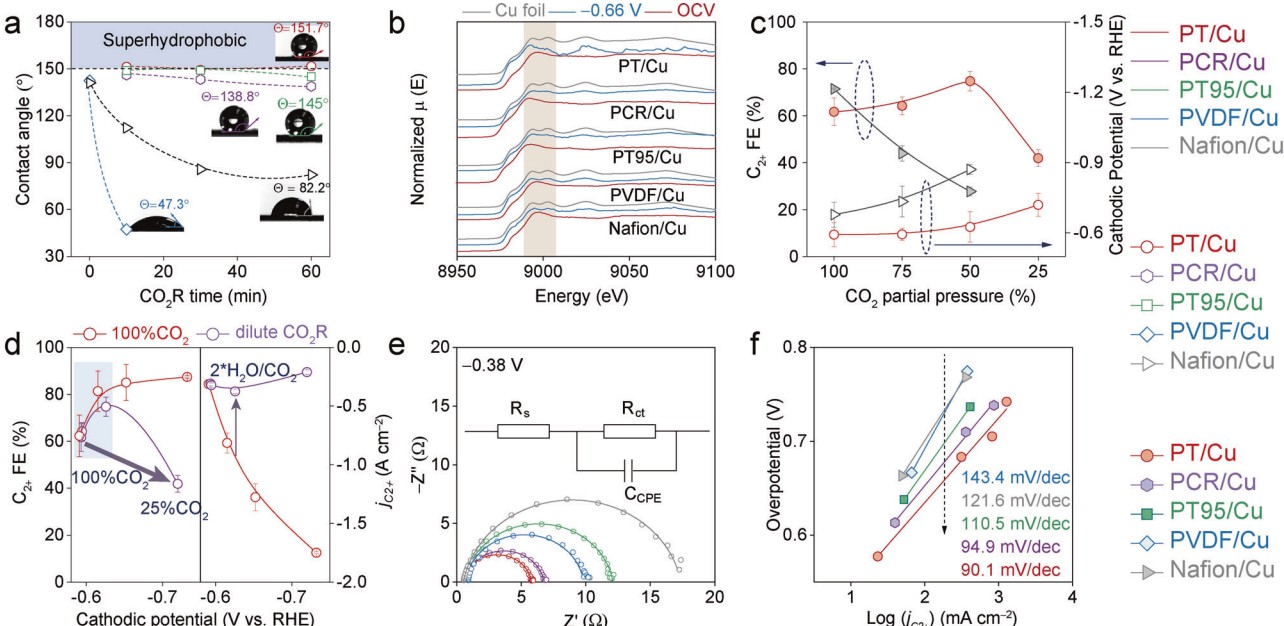

**Fig. 6 | Post CO₂R, in-situ measurements and reaction kinetics analysis. a** water contact angle for the five polymer/Cu GDEs after CO₂R at −0.5 A cm⁻². **b** in-situ XAS measurements. **c** CO₂R performance of C₂₊ selectivity and cathodic potentials on the function of CO₂ partial pressure at −0.5 A cm⁻², and **d** the influence of the local H₂O/CO₂ ratio (regulated by changing the CO₂ partial pressure) on CO₂R performances under identical cathodic potentials. The CO₂ partial pressure was adjusted by mixing CO₂ with Ar. A 50% CO₂ partial pressure correlates to a doubled H₂O/CO₂ ratio than that of 100% CO₂ concentration. **e** Nyquist plots of electrochemical impedance spectroscopy at −0.38 V vs. RHE in flow cell, the inset shows the equivalent circuit used for the curve fitting. The symbols in this figure are original

data, the solid lines are fitting data. **f** Tafel plots for the CO₂R to C₂₊ products for PT/Cu, PCR/Cu, PT95/Cu, PVDF/Cu and Nafion/Cu GDEs. The average equilibrium potential from CO₂R to C₂₊ products was deemed as 0.09 V vs. RHE. All the CO₂R tests here were conducted in a flow cell, and the electrolyte is 2 M KOH. For panels **c, d** and **f**, 100% *iR* correction was applied. Details regarding the *iR* correction can be found in the Method section. For the overpotential calculation in panel **f**, the average equilibrium potential for C₂₊ products were set as 0.09 V. The error bars in panel **c** and d represent standard deviations from at least three independent measurements. Relevant source data are provided as a Source Data file.

---

selected PT as a representative for the three relatively stable polymers and compared it with Nafion and PVDF. As shown in Supplementary Fig. 34, the potassium (K) concentration exhibits the same trend as the contact angle measurements. Specifically, K was observed throughout the entire GDEs of Nafion/Cu and PVDF/Cu. However, only limited K was found in PT/Cu, indicating negligible electrolyte flooding occurred. As a result, >150 h continuous CO₂R at −0.2 A cm⁻² was demonstrated on PT/Cu, with negligible changes in FEs for both CO₂R products and HER. In contrast, Nafion/Cu and PVDF/Cu exhibit poor CO₂R stability, for only 11 h and 4 h, respectively (Fig. 5g). Furthermore, PT/Cu maintained stability at even high current density, i.e. −1 A cm⁻², for ~10 h, indicating its impressive stability at large current densities (Supplementary Fig. 35). In contrast, Nafion/Cu displayed subpar CO₂R performance towards C₂₊ products initially at this high current density, achieving a C₂₊ FE of only 65%, which then decline rapidly. Overall, PT/Cu exhibits exceptional stability in maintaining the local H₂O/CO₂ for CO₂R, owing to its high chemical stability.

### Mechanistic investigations

The chemical state of the Cu catalyst during CO₂R was studied through *operando* XAFS spectroscopy with the same flow cell. As shown in Fig. 6b, the Cu K-edge XAFS spectra acquired for all five polymer/Cu GDEs suggest that Cu NPs were reduced to metallic state under real CO₂R conditions[58], indicating that the enhanced CO₂R activity of PT/Cu was not resulted from differences in chemical state. Besides, the morphological structures of the Cu NPs were measured before and post CO₂R for each GDEs, and no obvious changes were observed (Supplementary Fig. 36). We also do not expect the formation of any special active Cu-sites during the polymer coating. Therefore, the enhanced reaction CO₂R kinetics of PT/Cu is primarily attributed to the optimized microenvironment, especially the low local H₂O/CO₂ ratio.

The influence of local H₂O/CO₂ ratio was further investigated by tunning the CO₂ partial pressure ($P_{CO2}$) during CO₂R on PT/Cu. As shown in Fig. 6c, d, we observed an initially increased FE_{C2+} when reducing the $P_{CO2}$ (Supplementary Fig. 37, Supplementary Tables 13 and 14), until a very low $P_{CO2}$ of 25% was applied, likely due to the insufficient CO₂ supply. We believe that this initially increased FE_{C2+} was caused by the increased cathodic potentials (Fig. 6c)[52,53]. Similar CO₂ diffusion induced concentration overpotential changes were observed elewhere[59]. In contrast, the FE_{C2+} of Nafion/Cu dropped rapidly and monotonically with decreasing $P_{CO2}$ (Fig. 6c), indicating a severe CO₂ transfer limitation under low $P_{CO2}$, likely resulting from the unsatisfactory CO₂ uptake and the GDL flooding. On the other hand, we anticipate an increase of local H₂O/CO₂ ratio when reducing the $P_{CO2}$. To further understand the interplay between CO₂R performance and the local H₂O/CO₂ ratio, we compared the FE_{C2+} from CO₂R under both dilute and 100% CO₂ conditions, at identical applied cathodic potentials. As shown in Fig. 6d, at low $P_{CO2}$ we observed that the $j_{C2+}$ notably decreased (Fig. 6d, right figure) while the FE_{C2+} remain relatively constant (Fig. 6d, left figure), likely due to the increased H₂O/CO₂ ratio under the same cathodic potentials. These findings align with the simulation results in Fig. 3c–e, indicating that the local H₂O/CO₂ ratio substantially impacts CO₂R performance. It is crucial to note that our simulation results did not account for gas diffusion limitations caused by catalyst layer flooding, explaining why the CO₂R performance at 25% $P_{CO2}$ diverges from these simulations. The local H₂O/CO₂ ratio can be further regulated by adjusting the humidification of inlet CO₂. As presented in Supplementary Fig. 38a, an increase in the inlet CO₂ humidification results in a gradual rise in cell voltage with increasing temperature despite the changes are minor. As a result, this higher cell voltage led to an increase in C₂H₄ selectivity while reducing CO selectivity, as shown in Supplementary Fig. 38b. We anticipate that

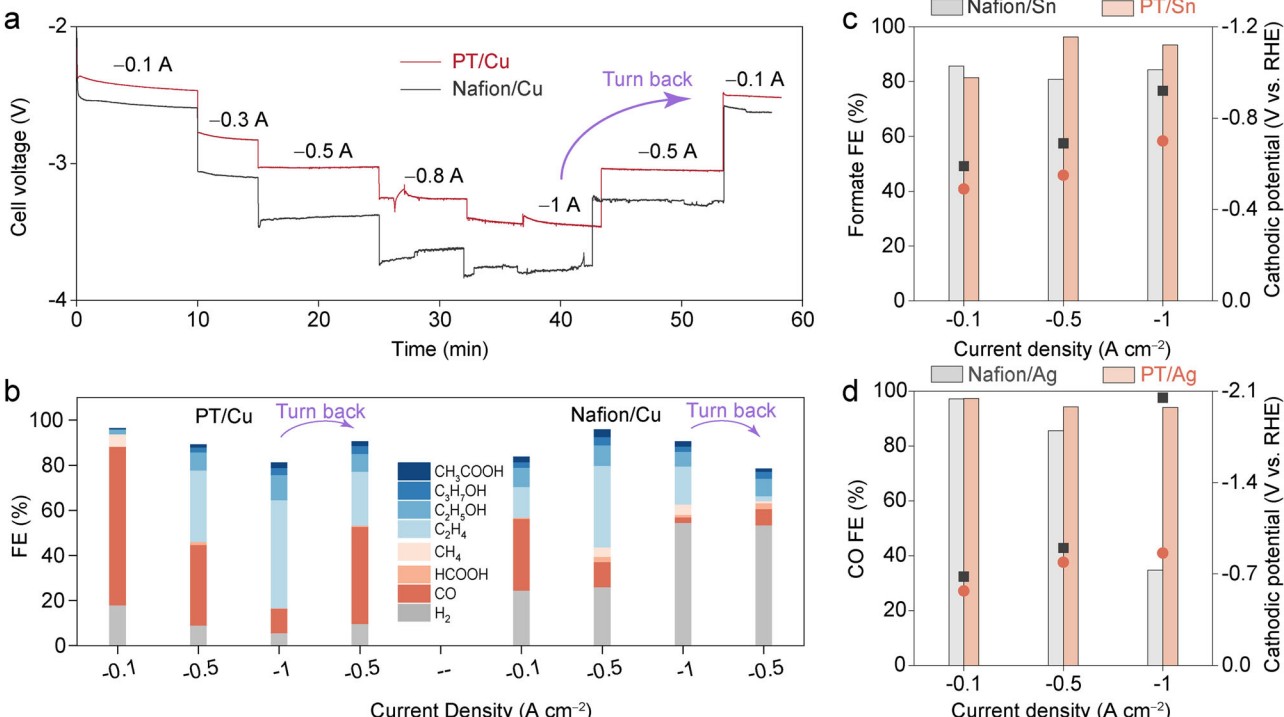

**Fig. 7 | PT coating for MEA reactor and other CO₂R catalysts. a** Full cell voltage on the function of the applied currents. **b** CO₂R products distribution of PT/Cu and Nafion/Cu. **c-d** CO₂R performance on (**c**) Sn NP and (**d**) Ag NP catalysts in a flow cell. For the MEA test, 1 M KOH was used as the anolyte with a flowrate of 3 sccm. The CO₂ flowrate in the cathode side is 24 sccm. Before introducing CO₂ into the MEA cell, it is passed through a sealed water container for preliminary humidification. For panel **a**, no *iR* correction was applied, for panels **c** and **d**, 100% *iR* correction was applied. Details regarding the *iR* correction can be found in the Method section. Relevant source data are provided as a Source Data file.

higher temperature would lead to a significant increase in CO₂ feed humidity, which on the other hand will result in increased H₂O/CO₂ ratio and consequently declined CO₂R activity as we observed in this study. Taken together, this phenomenon is in line with our conclusion and further confirms the critical role of the local H₂O/CO₂ ratio in regulating CO₂R performance.

Electrochemical impedance spectroscopy (EIS) was employed to probe the CO₂R kinetics. As shown in Fig. 6e, the Nyquist plots for all GDEs exhibit a single apparent semicircle. Based on the proposed equivalent circuit[60], we defined $R_{ct}$ and $R_s$ as the charge transfer resistance and reactor solution resistance, respectively, and conducted the curve fitting. As a result, we observed a significantly smaller $R_{ct}$ for PT/Cu compared to Nafion/Cu at −0.38 V. We attribute this to the enhanced conversion rate of CO₂ to CO on PT/Cu, resulting higher CO coverage and further promote the C₂₊ formation (Fig. 6e)[61,62]. This is evident by the higher CO partial current density observed for PT/Cu compared to the other four GDEs (Supplementary Fig. 28). Besides, a potential dependent EIS measurement was conducted for PT/Cu (Supplementary Fig. 39, Supplementary Table 15). The characteristic frequency of the semicircle shifts to higher values with increase in the cathodic potential, corresponding to an exponential decrease in $R_{ct}$[63,64]. On the other hand, the measured $R_s$ for each GDE were nearly identical since the same reactor configuration was employed (Supplementary Fig. 40). Note, the similar $R_s$ for these polymer/Cu GDEs also suggests that the ionic conductivities at the reaction interface are not the limiting factor for CO₂R on the GDEs, owing to the thin nature of the polymer coating, i.e., 2-5 nm (Fig. 4e–j). To further validate that the ions transfer across the thin polymer film is not the limiting factor in our system, we simulated the limiting current density under minimal water content in the PT polymer (extreme conditions). As shown in Supplementary Fig. 41, when considering a thin polymer layer (i.e. 3 nm) and only the diffusion of K⁺ (as K⁺ is the most abundant cation in

electrolyte), it was found that even with a very low water content and a polymer porosity decreased to 0.1, the corresponding limiting current density could still reach 29.2 A cm⁻². This value is substantially higher than the maximum current density achieved in our experiments (i.e. −2 A cm⁻²). This result indicates that the thin polymer coating is unlikely to hinder the ionic transfer across the reaction interface, at least within the current density range we measured. Next, Tafel plot-type analysis was conducted to investigate the kinetics of CO₂R. As shown in Fig. 6f, we observed a decrease in Tafel-slope value (from 143.4 mV dec⁻¹ to 90.1 mV dec⁻¹) in the order of PVDF>Nafion>PT95 > PCR > PT. The above observations imply that the polymers possess stable and low local H₂O/CO₂ ratio can faciliate the CO₂R kinetics, which increase the reaction rates towards C₂₊ products. Overall, these mechanistic investigations further confirmed the significance of an optimized local H₂O/CO₂ ratio, which we believe can serve as an effective descriptor for preparing active and stable GDEs for CO₂R.

### Effects of polymer coating thickness and electrolyte pH
We studied the effect of PT coating thickness on CO₂R performance. By varying the PT loading, Cu GDEs with different thicknesses of PT coatings were obtained (Supplementary Fig. 42 and Supplementary Table 16). As shown in Supplementary Fig. 43, the threshold for achieving the above promotion effect is the addition of 20 µL of PT solution during the catalyst ink preparation, corresponding to a coating thickness of ~2 nm (Supplementary Fig. 44). Beyond this threshold, the FE_C₂₊ remains relatively stable until a high PT loading (~80 µL) (Supplementary Table 17), likely due to the inefficient water uptake and/or electronic conductivity caused by the thick polymer-coating (Supplementary Fig. 45). The influence of thickness also aligns well with the simulation results (Supplementary Fig. 8). Our observations indicate that, with an optimal polymer coating thickness where conductivity was less significant, the thickness does not really

influence the effective diffusion coefficients of species within the domain. As a result, $CO_2$ reactants and other species within the electrolyte can quickly diffuse through the polymer layer, at least significantly faster than the reaction kinetics. Consequently, a moderate PT loading of 40 μL was used for the above and subsequent measurements.

CO$_2$R in acidic electrolyte has shown promise in improving the $CO_2$ utilization efficiency[65,66]. Thus, we also evaluated the promotional effect of PT for acidic CO$_2$R. As shown in Supplementary Fig. 46c, compared to Nafion/Cu, PT/Cu exhibited significantly lower overpotential (by at least >0.3 V) to achieve high current densities with high FE$_{C2+}$, confirming the enhanced acidic CO$_2$R activity of PT/Cu. Note that the issue of flooding is less severe in acidic CO$_2$R with Nafion/Cu catalysts (Supplementary Fig. 46b, Supplementary Table 18), likely due to the reduced formation of carbonate salts. Significantly, the maximum single-pass carbon efficiency of 46.3% (based on the CO$_2$R products) was achieved on PT/Cu at −1 A cm$^{-2}$ under acidic conditions (Supplementary Fig. 47).

### PT coatings in MEA reactor and other CO$_2$R catalysts

We proceeded to assess the CO$_2$R performance of PT/Cu in a membrane-electrode-assembly (MEA) reactor (Supplementary Fig. 48). As shown in Fig. 7a, the cell voltages of PT/Cu were significantly reduced compared to those of Nafion/Cu at broad current density ranges. Additionally, the selectivity of PT/Cu towards CO$_2$R products remained steady at ≥80%, even at high current density of −1.2 A cm$^{-2}$ (Fig. 7b, Supplementary Fig. 49, Supplementary Table 19). In contrast, Nafion/Cu exhibited much lower CO$_2$R selectivity, likely due to flooding occurring at the catalyst layer (i.e., up to 60% H$_2$ at −1 A cm$^{-2}$, Fig. 7b, right part). Compared to the state-of-the-art Cu-based MEA, PT/Cu shows notably improved CO$_2$R current (excluding HER) at a given cell voltage, indicating enhanced energy efficiency (Supplementary Fig. 50, Supplementary Table 20)[15,67–74]. Specifically, an encouraging EE of 21% was achieved at high current density of −1.2 A cm$^{-2}$, which could be further improved to >30% by increasing the anolyte alkalinity (Supplementary Figs. 51 and 52). When incorporating electrocatalysts with improved intrinsic activity compared to pristine Cu, and optimized anodic catalysts, we anticipate achieving further enhanced EE that meets the threshold for practical applications[69,72,75]. Moreover, similar enhancements in CO$_2$R selectivity and EE were observed for PT/Cu-based MEA using acid electrolyte (pH=1.5) (Supplementary Figs. 53 and 54).

Furthermore, we employed PT polymer for CO$_2$R on Sn nanoparticles (Sn NP) and Ag nanoparticles (Ag NP) for producing formate and CO, respectively. As shown in Fig. 7c, the FE$_{formate}$ of PT/Sn reaches >93% at high current density of −1 A cm$^{-2}$, at −0.7 V, which is ~220 mV lower than that of Nafion/Sn at the same current density (Supplementary Table 21). Similarly, PT/Ag achieved high FE of ~95% for CO formation at −1 A cm$^{-2}$ at −1 V. In contrast, Nafion/Ag experienced a rapid flooding under the same condition, resulting in low FE$_{CO}$ of 35% and high overpotential of > −2.0 V (Fig. 7d, Supplementary Table 22). Overall, these results indicate that our strategy has broad applicability for CO$_2$R across various reactor configurations and electrocatalysts.

## Discussion

In summary, we showcase a design principle for constructing catalyst layers with optimized local H$_2$O/CO$_2$ ratio for efficient CO$_2$R by employing thin polymer coatings to the catalyst surface. The key lies in selecting polymers with high porosity and CO$_2$ permeability, low water uptake, and robust chemical stability under CO$_2$R conditions. Based on these selection criteria, we identify the PT polymer as a candidate to validate our design strategy. Consequently, the PT modified Cu-GDE demonstrates significantly improved C$_{2+}$ products selectivity and energy efficiency at practical relevant current densities,

achieving high FE$_{C2+}$ exceeding 85% and cathodic EE of >51% at a high current density of −2 A cm$^{-2}$. Moreover, continuous CO$_2$R over 150 h is demonstrated at −0.2 A cm$^{-2}$ with negligible loss in both activity and selectivity. Furthermore, we show that this design principle is applicable to CO$_2$R with different reactor configurations (i.e. MEA) and electrocatalysts.

## Methods
### Chemicals and materials

The chemicals used for electrolytes and electrode preparation, including potassium hydroxide (99.99%), potassium sulfate (90%), potassium bicarbonate (≥99%), sulfuric acid (ACS reagent, 95%-98%), Nafion™ 117 solution (5% in a mixture of lower aliphatic alcohols and water), PTFE solution (60% dispersion in H$_2$O), Ethanol (ACS reagent), N,N-Dimethylformamide (suitable for HPLC, ≥99.9), Polytetrafluoroethylene particles (mean particle size 20 μm), Polyvinylidene fluoride and Copper nanoparticles (25 nm) were purchased from Sigma-Aldrich. Fully perfluorinated polymer (Poly[4,5-difluoro-2,2-bis(trifluoromethyl)−1,3-dioxole-co-1,3-Dioxane,2-(difluoromethylene)−4,4,5,5,6-pentafluoro-6-(trifluoromethyl)]: PT; Poly[2-(1,1-difluoroethyl)−2-ethyl-4,4,5,5,6-pentafluoro-6-(trifluoromethyl)−1,3-dioxane)]: PT95 and Poly[3,3,4,4-tetrafluoro-2-methyl-2-(1,1,2,2,3-pentafluoro-3-(trifluoromethoxy)butyl)−5,5-bis(perfluoroethyl)tetrahydrofuran]: PCR) and Fluorinert electronic liquid (FC-770) were purchased from Shanghai Puchun industrial Co., LTD. The anion-exchange membrane (Sustainion, X37-50-grade T), Sustainion ionomer (XC-2), and Nafion 117 membrane were purchased from Fuel Cell Store. All chemicals were used without further purification. The carbon paper (YLS-30T) used in flow cell and MEA was purchased from Suzhou Sinero Technology CO., LTD. and the carbon paper (P75T) used in H-cell was purchased from fuel cell store. Deionized water (18.2 MΩ) was used for the preparation of all electrolytes.

### Characterizations

A JEOL JSM-7610F scanning electron microscope was employed to acquire the energy-dispersive X-ray (EDX) mapping and field-emission scanning electron microscopy (FESEM) images. FETEM images were acquired using a JEOL JEM-2100 operating at 200 kV. X-ray photoelectron spectroscopy was conducted using a Kratos AXIS Ultra spectrometer, which features a monochromatized Al K X-ray source and a concentric hemispherical analyzer. X-ray diffraction (XRD) measurements were performed using a Shimadzu XRD-6000 diffractometer with a Cu Kα X-ray source, scanning at 5° per minute. The water and gas contact angles were measured using a goniometer (VCA Optima, AST Products Inc.) at 25 °C, with water droplets and air bubbles used for these measurements, respectively. Adsorption isotherms for N$_2$ and CO$_2$ were determined using a Micromeritics ASAP 2020 analyzer (3Flex Version 5.02). N$_2$ adsorption measurements were conducted under liquid nitrogen conditions, while CO$_2$ adsorption measurements were performed at room temperature. For the CO$_2$ adsorption isotherm study, approximately 100 mg of the corresponding polymer membranes first underwent vacuum degassing at 393.15 K for 6 hours to remove any impurities. After degassing, the samples were then cooled to room temperature, transferred to the analysis port, and measured within an absolute pressure range of 0-800 mmHg at room temperature. In-situ X-ray absorption spectroscopy (XAS) experiments were performed at the XAFCA beamline of the Singapore Synchrotron Light Source, utilizing fluorescence mode. The Cu K-edge X-ray absorption near-edge structure (XANES) spectra were collected and processed using Athena software. CO$_2$ permeance was measured using a custom-built gas permeation apparatus with an MKS instrument (Andover, MA, USA), following previously described procedures[76]. The thickness of the polymer membrane was measured using a Digimatic indicator (IDC-112b-5), and its density was determined by cutting a 1 cm diameter piece using abrasive tools and weighing its mass.

## Electrode preparation

The polymer solutions of PT95, PT, and PCR were prepared by dispersing the corresponding polymer powders into the FC-770 electronic liquid at concentrations of 1 wt% for PT and PCR and 2 wt% for PT95, respectively. The PVDF solution was prepared by mixing 3 wt% of PVDF powder with DMF. Catalyst inks were prepared by mixing 15 mg of Cu NPs with 1 mL of the same specific solvent used for preparing the polymer solutions, followed by adding a certain amount of polymer solutions to fabricate the corresponding polymer/Cu electrodes. Specifically, 40 μL of PT and PCR solutions and 60 μL of PT95 solution were employed for preparing the GDEs of PT/Cu, PCR/Cu, and PT95/Cu. DMF and ethanol were used as solvents to add 50 μL of PVDF solution and 75 μL of Nafion solution to fabricate the PVDF/Cu and Nafion/Cu electrodes, respectively. These procedures, with a certain loading of polymer in the Cu NPs catalyst inks, guarantees relatively constant polymer thickness on the catalyst surface. We also conducted preliminary estimation on the relevant polymer loadings (see details in Supplementary Table 16). After the preparation of the catalyst inks, they were sonicated for 30 minutes at room temperature to obtain homogenous solutions. These solutions were then sprayed onto a carbon fiber paper with the size of $2.5 \times 3 \, cm^2$ (YLS-30T). Then, the resulting carbon paper was dried under a vacuum for 5 hours at 80 °C to ensure no solvent residuals. Finally, the dried carbon electrodes were cut into pieces measuring $2.5 \times 1 \, cm^2$ and $1 \times 1 \, cm^2$ to serve as the cathodic electrodes in the flow cell and membrane electrode assembly (MEA) reactor, respectively. The catalyst loadings were estimated by weighing the carbon electrodes before and after the loading of the catalyst inks. In a typical experiment, we control the catalyst loading to be approximately $1 \, mg/cm^2$. For flow cell measurements, the back side of the carbon electrodes was pre-treated with PTFE solutions to boost its hydrophobicity to mitigate the potential electrolyte flooding. This procedure was done before the catalyst loading. Specifically, 2 mL of a 10% diluted PTFE solution was sprayed onto a carbon paper with a size of $10 \, cm^2$. Subsequently, the carbon paper underwent calcination at 350 °C in a muffle furnace for 30 minutes, with a heating rate of 5 °C/min, to establish a uniform and hydrophobic PTFE coating. For electrodes measured in H-cell, no PTFE treatment was made. However, the catalyst ink preparation process remained consistent with the procedure outlined above, except that the volume of solvents was raised from 1 to 3 mL. Subsequently, 200 μL of these inks were drop-cast onto a $1 \times 1 \, cm^2$ carbon paper (P75T).

## Cell assembly and electrochemical CO$_2$ reduction measurements

**Flow-cell assembly.** The flow cell, depicted in Supplementary Fig. 5, was purchased from GaossUnion. It comprises a gas chamber, a catholyte chamber (housing an Ag/AgCl reference electrode), an anolyte chamber, and sealing plates. After assembling, the catholyte and anolyte chambers are separated by a proton exchange membrane (Nafion 117). The working electrode is positioned between the gas chamber and the catholyte chamber, while the counter electrode (IrO$_2$-loaded Ti mesh) is placed between the anolyte chamber and the anodic plate[77]. During the CO$_2$R measurements, continuous CO$_2$ gas flow (24 sccm) regulated by a digital mass flow controller (Sevenstar, MFC CS200-A) was supplied to the gas chamber constantly. The electrolytes circulated through the catholyte and anolyte chambers were controlled by two peristaltic pumps at flow rates of 3 mL min⁻¹ and 5 mL min⁻¹, respectively. In the CO$_2$ partial pressure dependent CO$_2$R experiment, the CO$_2$R electrolysis was carried out at the current density of −0.5 A cm² using 2 M KOH as the electrolyte. Besides, the CO$_2$ partial pressure was varied by mixing CO$_2$ with Ar. Regarding the use of a proton exchange membrane, make sure it is completely hydrated. Specifically, prior to using, submerge the membrane in deionized water for at least an hour. After that, boil the membrane in deionized water for an hour to get rid of any contaminants on its surface. Subsequently, immerse the membrane in a 5% hydrogen peroxide (H$_2$O$_2$) solution for

another hour at 80 °C, followed by thoroughly rinse with deionized water. Then, boil the membrane in a 0.5 M sulfuric acid (H$_2$SO$_4$) solution for one hour at 80 °C to initiate membrane protonation. Lastly, to ensure all chemicals are removed, boil the membrane in deionized water for another hour. Store the treated membrane in deionized water to keep it hydrated.

**H-cell assembly.** The H-cell used in our study, purchased from GaossUnion, consists of airtight dual compartments. In the anodic compartment, a Pt foil counter electrode (area: $1 \times 2 \, cm^2$) is employed, while the cathodic compartment contains an Ag/AgCl reference electrode and the working electrode. These two compartments were separated by a piece of Nafion 117 membrane, and each compartment contains 20 mL of electrolyte (0.5 M KHCO$_3$). Prior to CO$_2$R measurements, the electrolyte was saturated with 99.995% CO$_2$ for at least 30 minutes with a CO$_2$ flow rate of 24 sccm to mainain its saturation. To evaluate the electrochemically active surface area, all five polymer/Cu electrodes were initially activated by applying a bias at −10 mA cm⁻² for 30 minutes. For the CV testing, a potential range of 0.4-0.5 V vs. Ag/AgCl, and scan rate ranging from 10 to 120 mV/s were used. At least ten CV cycles were collected at each scan rate.

**Membrane electrode assembly (MEA).** Full cell measurements were peroformed using a commercial MEA electrolyzer (Suzhou Sinero Technology CO., LTD, 1 cm² active area). The scheme of the MEA is shown in Supplementary Fig. 48. This MEA consisted of a cathode electrode, anion-exchange membrane (Sustainion, X37-50-grade T) and anode electrode (IrO$_2$-Ti mesh[77]). The electrodes were then respectively mounted on their flow fields, seperated by the anion-exchange membrane and assembled in the MEA electrolyzer. In the cathode, the CO$_2$ flowrate was set as 24 sccm. Before introducing CO$_2$ into the MEA cell, it is passed through a sealed water container for preliminary humidification. In the anode, 1 M KOH served as the electrolyte and was circulated into the anode chamber at a rate of 3 sccm using a peristaltic pump. For testing in acidic conditions, the electrolyte was switched to 0.6 M K$_2$SO$_4$, with H$_2$SO$_4$ added to adjust the pH to 1.5. The liquid products were collected from both the anode and cathode. To collect the liquid products in the cathode, a cold trap was used to condense the outlet gas flow. In a typical measurement, the current density was progressively increased from −0.1 to −1 A cm⁻², then reversed back to −0.1 A cm⁻². To regulate the humidification of the inlet CO$_2$ in the MEA test, a sealed water reservoir was heated in a water bath, with the temparature of the water bath controlled between 40 °C and 70 °C. The Sustainion membrane was immersed in 1 M KOH for 24 h before use to activate it.

The resistance of the flow cell was measured using the potentiostatic electrochemical impedance spectroscopy (PEIS) method by scanning from 1 MHz to 1 Hz before and after the electrolysis process. In EIS results, the intersection of the curve with x-axis represents the solution resistance. For the flow cell and H-cell, all the potential readings were measured against Ag/AgCl (reference potential: −0.197 V vs. SHE), and then converted to RHE using E (versus RHE) = E (versus Ag/AgCl) + 0.193 V + 0.0591 × pH, with necessary $iR$ compensation: 85% $iR$ compensation for polarization curves and 100% $iR$ compensation for other tests. The 85% $iR$ compensation for polarization curves was applied because one solution resistance value was used for all potentials. We speculate that the solution resistance slightly changes due to the local heating, particularly in regions with high current density. Therefore, to avoid potential overcompensation, 85% $iR$ compensation was used. For other tests, solution resistance was measured at each current density before and after electrolysis, allowing for 100% $iR$ compensation. In MEA, the full cell voltages were obtained without $iR$ correction.

The Ag/AgCl reference was calibrated using a reversible hydrogen electrode (RHE) under controlled conditions. The calibration was

performed in a high-purity $H_2$-saturated electrolyte (0.5 M $H_2SO_4$, pH=0) with a Pt wire as the working electrode, a graphite rod as the counter electrode, and the Ag/AgCl electrode as the reference. CVs were run at a scan speed of $1\,mV\,s^{-1}$. The average value of the two potentials at which the current equals 0 was recognized as the thermodynamic potential for the hydrogen evolution reaction (E = 0 V vs. RHE). The measured value for RHE is −0.197 V vs. Ag/AgCl, therefore the calibrated potential for E (vs. Ag/AgCl) is 0.197 V.

All of the electrochemical tests were performed with a bio-Logic VMP3 multichannel potentiostat/galvanostat that had a built-in EIS analysis. The EIS fitting was conducted with ZView software. For the flow cell tests, 2 M KOH was used as an alkaline electrolyte, and the acid electrolyte was composed of 0.6 M $K_2SO_4$ and $H_2SO_4$ (used to adjust the pH to 1.5). In the H-Cell, the electrolyte was 0.5 M $KHCO_3$. For the MEA, 1 M KOH was used as the alkaline electrolyte and the acid electrolyte was the same as the electrolyte used in the flow cell tests. To prepare the above electrolyte, a calculated amount of salt (KOH, $K_2SO_4$, and $KHCO_3$) was weighed and dissolved in 500 mL of ultra-pure water, which was contained in a sealed Polypropylene (PP) bottle. The corresponding solution pH was measured using a benchtop pH Meter (METTLER TOLEDO S400-Std-Kit). The pH values for 1 M KOH, 2 M KOH and 0.5 M $KHCO_3$ ($CO_2$ saturated) are 14.01 ± 0.04, 14.32 ± 0.06, and 8.21 ± 0.03, respectively.

The actual outlet gas flow rate of the test cell was measured by a bubble flowmeter. Gaseous reduction products were analyzed using on-line gas chromatograph (GC, Shimadzu 2014). The gas flow rates were measured at the inlet of GC (after the outlet of the $CO_2R$ reactor) for gaseous product quantification in each experiment with a bubble flowmeter. Liquid products were identified by $^1H$ NMR spectrum (Bruker 400 MHZ system) with Phenol and DMSO as the internal standard. The Faradic efficiency of the gas product was calculated on the basis of the following equation:

$$\text{Faradic efficiency} = \frac{i_x}{i_{total}} = \frac{n_x v_{gas} c_x F}{i_{total} V_m} \qquad (1)$$

Where $i_x$ is the partial current density of product $x$, $i_{total}$ is the total current density, $n_x$ represents the number of electrons transferred towards the formation of product $x$, $v_{gas}$ is the $CO_2$ flow rate (sccm), $c_x$ represents the concentration of product $x$ detected by gas chromatography (ppm), $F$ is the Faradic constant (96,485 C $mol^{-1}$), and $V_m$ is the unit molar volume, which is 24.51 $mol^{-1}$ at room temperature (298.15 K).

The cathodic energy efficiency for each product was calculated as follows:

$$EE_x = \frac{E_x}{E} * FE_x \qquad (2)$$

For the flow cell, $E_x$ is the equilibrium full cell potential $\left(E^0_{CO2/x} - E^0_{water\ oxidation}\right)$, where $E^0_{CO2/x}$ is the thermodynamic potentials (vs. RHE) of product $x$. $E^0_{water\ oxidation}$ is 1.23 V. $FE_x$ is the measured FE for product $x$. $E$ is the applied full cell potential $\left(E^{applied} - E^{assumed}_{water\ oxidation}\right)$. Here, the $E^{assumed}_{water\ oxidation}$ is the anodic water oxidation potentials assuiming no overpotential. $E^{applied}$ is the average applied cathodic potential (with Ohmic loss correction, V vs. RHE). The reference electrode used in our measurements is an Ag/AgCl electrode, with a calibrated potential of 0.197 V vs. RHE. In the MEA, $E$ represents the tested full cell voltage without iR correction, and $E_x$ is the equilibrium full cell potential as described above.

The $CO_2$ SPCE towards each product was determined using the following equation at ambient atomosphere:

$$SPCE = \frac{(j_{x*60s})/(n*F)}{(v*1\,min)/V_m} \qquad (3)$$

Where $j_x$ is the partial current density (A) of product $x$, and $n$ is the electron transfer for the formation of product $x$, $v$ is the $CO_2$ flow rate and $V_m$ equals to 24.51 $mol^{-1}$.

## Water uptake ability test
The water uptake ability of the five polymers was evaluated using their membrane forms, as shown in supplementary Fig. 14. Unlike the polymer powder, the polymer membrane closely resembles the state of the polymer on the catalyst surface. Therefore, assessing the water uptake ability in the membrane form provides an approximation of the water uptake under practical conditions. To manufacture the membranes, 3 mL of each polymer solution was drop-cast onto a PTFE film, followed by heating to 80 °C. As the solvents gradually evaporated, the polymer membranes were formed. To assess the water uptake ability of each polymer membrane, the polymers were first dried at 80 °C for 24 h to remove adsorbed water and then weighed (donated as $W_d$). Subsequently, the polymers were immersed in ultrapure water for 24 h, after which their masses were determined again (donated as $W_h$). The process of oven drying and rehydration was repeated several times, and the weights of the hydrated and dried polymers were recorded in supplementary Table 4. The water uptake ability was calculated by the following equation:

$$\text{Water uptake ability}(\%) = \frac{W_h - W_d}{W_d} * 100\% \qquad (4)$$

All the calculated water uptake values were recorded in supplementary Table 5.

## In situ X-ray absorption spectroscopy (XAS)
In situ X-ray absorption spectroscopies of Polymer/Cu catalysts at the Cu K-edges were measured at the XAFCA beamline of Singapore Synchrotron Light Source using a fluorescence mode. The cell used for testing was the same as our flow cell. In situ spectra of polymer/Cu catalysts were collected from the backside of the carbon paper electrode. The electrochemical cell operation was controlled by a CHI760E potentiostat. A 2 M KOH solution served as the electrolyte and was purged into both the cathode and anode chambers of the flow cell using a double channel Peristaltic pump throughout the experiment. To avoid erosion by the alkaline electrolyte, the samples in the open-circuit voltage (OCV) state were tested without electrolyte flow.

## Data availability
Source data are provided with this paper and are available from the corresponding authors upon request. Source data are provided with this paper.

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

## Acknowledgements

We would like to acknowledge the support by The Basic Science Center Program for Ordered Energy Conversion of the National Natural Science Foundation of China (No.52488201). We also acknowledge the National University of Singapore the Ministry of Education for their financial support through the grants of A-0009176-02-00 and A-0009176-03-00, A*STAR (Agency for Science, Technology, and Research) under its LCERFI program (Award No U2102d2002), Centre for Hydrogen Innovations at NUS (CHI-P2022-06). L. Wang would also like to acknowledge the support of the National Research Foundation (NRF) Singapore under NRF Fellowships (NRF-NRFF13-2021-0007). H. Qiu acknowledges the support from the China Scholarship Council.

## Author contributions

L.W. supervised the project. L.W and J.C. conceived the idea. J.C. designed and performed the experiments. H.Q. and Y.Z. proposed the COMSOL model and H.Q. carried out the COMOSOL simulations. H.Y. and S.Xi helped with the XAS experiments. L. F. and L. C helped with the MEA assemble. Z.L. and G.Z. helped with the BET and XPS measurements. L.W., J.C., H.Q., J.C., Y.L., and L.G. contributed to the data interpretation and wrote the manuscript. All the authors commented on the manuscript.

## Competing interests

The authors declare no competing interests.
