## [Peer Review File · Nature Communications]

REVIEWER COMMENTS

Reviewer #1 (Remarks to the Author):

The authors have completely and satisfactorily answered most of my concerns. I agree with the authors that their new setup of the multiphysics model is more physical and a better representation of the experiment. My one concern before publication is that the authors need to study the effect of pore size in their modelling. In their response, the authors changed boundary conditions when considering different pore sizes, but we could expect pores that are fully wetted, of differing sizes, between 10nm and up to 100+nm.

Reviewer #2 (Remarks to the Author):

The authors have made substantial efforts to address the previously raised review questions. After the revision with new experimental data and clarifications, the quality of the manuscript has been improved. Therefore, I think this manuscript can be considered for publication in Nature Communications, but there are some minor points that the authors should still address, as listed below.

(1) Page 25, Line 2: "Flow-cell assembly: The flow cell was purchased from GaossUnion, as illustrated in Supplementary Fig. 5". I think it should be Supplementary Fig. 7.

(2) The Methods section indicated "a catholyte chamber (where a Ag/AgCl reference electrode locates)", but for the flow cell in Supplementary Fig. 7, the reference electrode was located at the anode chamber side next to the PEM. The authors should double check the schematic drawing.

(3) For the polarization curves in Figure 5a, the current density for the PT/Cu electrode increased so rapidly with the overpotential. I am curious what the Tafel slope will be if the authors draw a Tafel plot from the polarization curve. Similarly, for the data in Figure 7d, it seems that there is a minimal change of the potential from -0.5 to -1 A cm⁻², raising a similar question that why the current density increased so rapidly with the overpotential.

Reviewer #3 (Remarks to the Author):

Numerous studies have explored hydrophobic coating of cathodes for CO₂ reduction, yet a comprehensive understanding of the detailed mechanisms, supported by experimental evidence, remains lacking. This work presents expanded experimental findings augmented by computational simulations. Through analysis, the authors investigate the combined effects of hydrophobicity, gas permeability, water uptake, and polymer porosity on local H₂O/CO₂ ratios and CO₂ reduction performance. The simulation of reaction dynamics aligns closely with experimental observations. The reviewer has provided detailed feedback, resulting in manuscript enhancements. Overall, the paper is now deemed suitable for publication in Nature Communications.

Comments from Reviewer 1 and revisions made accordingly:

The authors have completely and satisfactorily answered most of my concerns. I agree with the authors that their new setup of the Multiphysics model is more physical and a better representation of the experiment. My one concern before publication is that the authors need to study the effect of pore size in their modelling. In their response, the authors changed boundary conditions when considering different pore sizes, but we could expect pores that are fully wetted, of differing sizes, between 10 nm and up to 100+ nm.

Response:

We thank the reviewer for this insightful comment and great suggestion. In response, we carefully revisited our previous modeling and found that altering the pore size of the catalyst layer can indeed affect the effective diffusion coefficient of the CO₂ gas by influencing its Knudsen diffusivity within the simulation area (*Phys Chem Chem Phys* **20**, 16973-16984 (2018)):

$$D_{\text{CO}_2,\text{g}}^{\text{K}} = \frac{2d_{\text{CL}}}{3} \sqrt{\frac{8RT}{\pi M_{\text{CO}_2}}}$$

Accordingly, we assessed the effect of the catalyst layer pore size, ranging from 10 nm to 100 nm, on the CO₂R performance of our system. The pores are assumed to be fully wetted. First, we simulated the local CO₂ concentration at the reaction interface as a function of the pore size. As shown in Fig. R1a, we observed a gradual rise in the average local CO₂ concentration as the pore size increased, however, with a relatively modest increase of approximately 1.5 mM when the pore size increased tenfold, from 10 nm to 100 nm. Consequently, the cathodic current density shows a minor decrease as the pore size decreases from 100 nm to 40 nm (Fig. R1b). Notably, we observed a sharp decline in the current density when we further reduced the pore size to 10 nm, indicating that significant CO₂ mass transportation limitation occurred at this scale. Furthermore, the changes in both the local pH (Fig. R1c) and cathodic potential (Fig. R1d) as a function of pore size show a similar trend to what was observed for the cathodic current density, indicating that the effect of the pore sizes on the CO₂R is minimal unless a small pore size of < 10 nm predominates in the catalyst layer.

Fig. R1. (a) Comparison of (a) average local CO₂ concentration, (b) simulated polarization curves, (c) average local pH and (d) cathodic potential versus RHE for GDEs with catalyst layers of varying pore sizes. In all cases, the porosity of the catalyst layer is maintained at 0.5, the H₂O/CO₂ ratio in the polymer is set at 0.1, and the applied potential at the catalyst is fixed at -1.426 V.

On the other hand, we used the method of airbrushing to prepare the catalyst layers in GDEs for all our measurements. Due to the high evaporation rate of the solvent (*i.e.*, ethanol and FC-770), as well as the micrometer scale and high surface-to-volume ratio aerosol mists (catalyst ink) induced by the pressurized gas, the resulting catalyst layers are highly porous, with well-scattered nanoparticles before they can settle down into the microporous layer. This porous structure has been demonstrated in a previous report (*Joule*, **4**, 1104-1120 (2020)). As illustrated in Fig. R2, the catalyst layer prepared through airbrushing was four times thicker and more porous than the catalyst layers prepared by drop-casting and hand-painting. Similar porous catalyst structure, achieved through the airbrushing method, was also observed in another report (*Fuel Cells*, **10**, 966-972 (2010)). In this work, an average pore size of approximately 230 nm was determined by Focused Ion Beam (FIB).

Fig. R2. Physical Characterization of Catalyst Layers Assembled by Cu_2O NPs via Various Deposition Techniques. (C–H) Cross-sectional SEM images and their energy-dispersive X-ray spectroscopy (EDS) elemental mappings of Cu for catalyst layers prepared by dropcasting ([C] and [D]), hand-painting ([E] and [F]), and airbrushing ([G] and [H]). The high porosity of the airbrushed sample is evident from the presence of macropores as denoted by the arrows in (G). (Ref. *Joule*, **4**, 1104-1120 (2020)).

Fig. R3. SEM images of polymer/Cu GDEs. The red arrows indicate typical pores in the catalyst layer.

Moreover, we also optimized the spray-coating parameters of our airbrush (heat temperature, discharging rate, *etc.*) to obtain highly porous catalyst layers according to protocols reported previously (*Joule*, **4**, 1104-1120 (2020)). Therefore, the catalyst layers in our study are expected to be highly porous, with pore sizes substantially larger than 10 nm. This is supported by the SEM images of the polymer/Cu GDEs prepared in our studies. As shown in Fig. R3, porous structures were observed for all the GDEs, with most measured pore sizes larger than 200 nm in PT/Cu (pores in other polymer/Cu GDEs are even larger). Although achieving even and precise control over the pore size of the catalyst layer is extremely challenging practically, we tried to keep our catalyst layer preparation protocols consistent for all the polymer/Cu GDEs in this study, to ensure relatively large pore sizes based on ours and documented experiences. Based on these considerations, we set the pore diameters within the catalyst layer in the current simulations to be a constant value of 200 nm to achieve reasonable comparisons. Consequently, the reasonably good match between the simulation results and our experimental observations confirms that the real pore size within the catalyst layers in our studies is substantially larger than 10 nm.

Revision made: The above Fig. R1 and Fig. R3 have been added as Supplementary Fig 11 and Supplementary Fig. 12, respectively. The relevant discussion has been added to the revised manuscript on page 8. “As the pore size of the catalyst layer can affect the effective diffusion of CO₂ gas⁴⁷, we explored the changes in CO₂R performance when the catalyst layer pore size ranged from 10 nm to 100 nm, assuming the pores are fully

wetted³⁵ (Supplementary Fig. 11). First, we simulated the local CO₂ concentration as the pore size increased, however, with a relatively modest increase of approximately 1.5 mM when the pore size increased tenfold, from 10 nm to 100 nm. Consequently, the cathodic current density shows a minor decrease as the pore size decreases from 100 nm to 40 nm. Notably, we observed a sharp decline in the current density when we further reduced the pore size to 10 nm, indicating that significant CO₂ mass transportation limitation occurred at this scale. Furthermore, the changes in both the local pH (Supplementary Fig. 11c) and cathodic potential (Supplementary Fig. 11d) as a function of pore size show a similar trend to what was observed for the cathodic current density, indicating that the effect of the pore sizes on the CO₂R is minimal unless a small pore size of < 10 nm predominates in the catalyst layer.”

Additionally, a detailed Note has been added to the Supplementary Information under Supplementary Fig. 11.

“We used an air-brushing method to prepare the GDEs. Due to the high evaporation rate of the solvent (*i. e.*, ethanol and FC-770), as well as the micrometer scale and high surface-to-volume ratio aerosol mists (catalyst ink) induced by the pressurized gas, the resulting catalyst layers are porous, with well scattered nanoparticles before they can settle down into the microporous layer. This porous structure has been demonstrated in a previous report¹⁴, where the airbrushed catalyst layer was four times thicker and more porous than the drop-casted and hand-painted layers. Similar porous catalyst structure, achieved through the airbrushing method, was also observed in another report¹⁵. In this work, an average pore size of approximately 230 nm was determined by Focused Ion Beam (FIB).

We also checked our catalyst layer by the SEM images of these polymer/Cu GDEs, As illustrated in Supplementary Fig. 12, porous structures were observed for all the prepared GDEs, with most measured pore sizes larger than 200 nm in PT/Cu (pores in other polymer/Cu GDEs are even larger). Although achieving even and precise control over the pore size of the catalyst layer is extremely challenging practically, we tried to keep our catalyst layer preparation protocols consistent for all the polymer/Cu GDEs in this study, to ensure relatively large pore sizes based on ours and documented experiences. Based on these considerations, we set the pore diameters within the catalyst layer in the current simulations to be a constant value of 200 nm to achieve reasonable comparisons. Consequently, the reasonably good match between the simulation results and our experimental observations confirms that the real pore size within the catalyst layers in our studies is substantially larger than 10 nm.”

Comments from Reviewer 2 and revisions made accordingly:

Reviewer #2 (Remarks to the Author):

The authors have made substantial efforts to address the previously raised review questions. After the revision with new experimental data and clarifications, the quality of the manuscript has been improved. Therefore, I think this manuscript can be considered for publication in Nature Communications, but there are some minor points that the authors should still address, as listed below.

We are very grateful for the recognition and in-depth comments/suggestions from the Reviewer. Below is our point-by-point response to the raised comments. Additionally, we have incorporated the necessary modifications into the revised Manuscript and Supporting Information, highlighting them with yellow background.

(1) Page 25, Line 2: “Flow-cell assembly: The flow cell was purchased from GaossUnion, as illustrated in Supplementary Fig. 5”. I think it should be Supplementary Fig. 7.

Response: We thank the reviewer for the thorough review. Accordingly, we have corrected the typo, changing “Supplementary Fig. 5” to “Supplementary Fig. 7” on page 25, Line 2, in the Revised Manuscript. We have also carefully reviewed the manuscript to avoid similar mistakes.

(2) The Methods section indicated “a catholyte chamber (where a Ag/AgCl reference electrode locates)”, but for the flow cell in Supplementary Fig. 7, the reference electrode was located at the anode chamber side next to the PEM. The authors should double check the schematic drawing.

Response: We thank the reviewer for the careful review and apologize for the incorrect labeling of electrodes in the flow-cell schematic. We have corrected the label accordingly. Additionally, to avoid any potential copyright issues, we redrew the schematic ourselves and replaced the previous version.

Fig. R4. Flow-cell schematic. Reactant gas is fed through the back of a gas diffusion-electrode.

Revision made:

The above Figure has been updated as Supplementary Fig. 7.

(3) For the polarization curves in Figure 5a, the current density for the PT/Cu electrode increased so rapidly with the overpotential. I am curious what the Tafel slope will be if the authors draw a Tafel plot from the polarization curve. Similarly, for the data in Figure 7d, it seems that there is a minimal change of the potential from -0.5 to -1 A cm^{-2} , raising a similar question that why the current density increased so rapidly with the overpotential.

Response: We thank the reviewer for raising this important question. As suggested by the reviewer, we plotted the Tafel slopes for the polarization curves in Figure 5a, as depicted in Fig. R5a. As the selectivity for each specific products are not available for these polarization curves, we assumed an average value of approximately 0 V (vs. RHE) for the common products (CO, C₂H₄, C₂H₅OH) observed in our system as the equilibrium potential for electroreduction of CO₂ to estimate the overpotentials. As

shown in Fig. R5a, similar Tafel slopes were observed for all five polymer/GDEs at relatively small current densities, where CO₂ mass transportation limitations are considered minimal. Specifically, in a typical case, the Tafel slope observed is slightly larger than 120 mV/dec, which is an anticipated value for a common PCET process. At high current densities, these Tafel slopes increased due to the CO₂ mass transportation limitations mostly induced by electrolyte flooding. Nevertheless, the increases in Tafel slope for PT/Cu electrode is substantially smaller than those observed for Nafion/Cu. This result confirms that PT could mitigate the flooding issues at the reaction interface. Additionally, we speculate that solution resistance may change due to the local heating, particularly in regions with high current density on these polarization curves. Hence, to avoid potential *iR* overcompensation, we plot the Tafel slope with 80% *iR* compensation, as shown in Fig. R5b. The figure shows that the Tafel slopes have minimal differences compared to those with 100% compensation, indicating that solution resistance has minimal influence on the Tafel slopes. Subsequently, the corresponding polarization curves are plotted in Fig. R5c.

We also assessed the Tafel slopes for CO₂R products derived from the Chronoamperometry test (Supplementary Fig. S22). As we tested the solution resistance at each current density before and after electrolysis, we applied 100% *iR* compensation. As shown in Fig. R5d, the Tafel slopes for all polymer/Cu GDEs closely resemble those derived from the polarization curves with 80% *iR* compensation. Therefore, we believe that these Tafel slopes are reasonable.

Regarding the small change of potential in Figure 7d, we also plotted the Tafel-type slope (as only three data points were used for this analysis, we define these slopes are Tafel-type slope). As shown in Fig. R5e, the Tafel-type slope value for Nafion/Ag is much higher than 120 mV/dec, which is understandable given that these data were obtained from regions of relatively high current density within a flow cell, often subject to mass transport limitations and electrolyte flooding to some extent. In contrast, the PT/Ag exhibits a substantially smaller Tafel-type slope compared to Nafion/Ag. This observation supports our conclusion that the PT polymer effectively enhances local CO₂ diffusion and mitigates flooding, particularly under conditions of high current densities.

Taken together, we believe the Tafel slopes observed in our study (approximately 120 mV/dec) are reasonable for a typical electrochemical process, and the introduction of PT indeed helps preventing the electrolyte flooding and maintaining stable Tafel slopes.

Fig. R5. Tafel slope from polarization curves with (a) 100% iR compensation and (b) 80% iR compensation; (c) polarization curves with 80% iR compensation; Tafel slopes derived from (d) Chronoamperometry test (Supplementary Fig. S22) and (e) Figure 7d.

Revision made: Figure 5a in the revised manuscript has been replaced by Fig. R5c above. The caption of Fig. 5a now includes “with 80% iR compensation”. Additionally, a description regarding iR compensation has been added to the Methods section on page 26. “and then converted to RHE with necessary iR compensation: 80% iR compensation for polarization curves and 100% iR compensation for other tests. The 80% iR compensation for polarization curves was applied because one solution resistance value was used for all potentials. We speculate that the solution resistance slightly changes due to the local heating, particularly in regions with high current density. Therefore, to avoid potential overcompensation, 80% iR compensation was used. For other tests, solution resistance was measured at each current density before and after electrolysis, allowing for 100% iR compensation.”

Comments from Reviewer 3 and revisions made accordingly:

Reviewer #3 (Remarks to the Author):

Numerous studies have explored hydrophobic coating of cathodes for CO₂ reduction, yet a comprehensive understanding of the detailed mechanisms, supported by experimental evidence, remains lacking. This work presents expanded experimental findings augmented by computational simulations. Through analysis, the authors investigate the combined effects of hydrophobicity, gas permeability, water uptake, and polymer porosity on local H₂O/CO₂ ratios and CO₂ reduction performance. The simulation of reaction dynamics aligns closely with experimental observations. The reviewer has provided detailed feedback, resulting in manuscript enhancements. Overall, the paper is now deemed suitable for publication in Nature Communications.

We greatly appreciate the reviewer's positive comments and recognition.

REVIEWERS' COMMENTS

Reviewer #1 (Remarks to the Author):

The additional study regarding the pore sizing is satisfactory and gives more confidence in the numerical results. As such, all of my concerns are addressed and the manuscript is suitable for publication in Nature Communications.

Reviewer #2 (Remarks to the Author):

The authors have addressed the review comments in a reasonable way and further improved the quality of the manuscript. Therefore, I would like to recommend its publication in Nature Communications.